# EXPLORING WEIGHT BALANCING ON LONG-TAILED RECOGNITION PROBLEM

**Naoya Hasegawa & Issei Sato**
The University of Tokyo
{hasegawa-naoya410, sato}@g.ecc.u-tokyo.ac.jp

## ABSTRACT

Recognition problems in long-tailed data, in which the sample size per class is heavily skewed, have gained importance because the distribution of the sample size per class in a dataset is generally exponential unless the sample size is intentionally adjusted. Various methods have been devised to address these problems. Recently, weight balancing, which combines well-known classical regularization techniques with two-stage training, has been proposed. Despite its simplicity, it is known for its high performance compared with existing methods devised in various ways. However, there is a lack of understanding as to why this method is effective for long-tailed data. In this study, we analyze weight balancing by focusing on neural collapse and the cone effect at each training stage and found that it can be decomposed into an increase in Fisher's discriminant ratio of the feature extractor caused by weight decay and cross entropy loss and implicit logit adjustment caused by weight decay and class-balanced loss. Our analysis enables the training method to be further simplified by reducing the number of training stages to one while increasing accuracy. Code is available at https://github.com/HN410/Exploring-Weight-Balancing-on-Long-Tailed-Recognition-Problem.

## 1 INTRODUCTION

Datasets with an equal number of samples per class, such as MNIST (Lecun et al., 1998) and CIFAR100 (Krizhevsky, 2009), are often used, when we evaluate classification models and training methods in machine learning. However, it is empirically known that the size distribution in the real world often shows a type of exponential distribution called Pareto distribution (Reed, 2001), and the same is true for the number of per-class samples in classification problems (Li et al., 2017; Spain & Perona, 2007). Such distributions are called long-tailed data due to the shape of the distribution since some classes (head classes) are often sampled and many others (tail classes) are not sampled very often. Long-tailed recognition (LTR) is used to attempt to improve the accuracy of classification models on uniform distribution when training data shows such a distribution. There is a problem in LTR that the head classes have large sample size; thus, the output is biased toward them. This reduces the overall and tail class accuracy because tail classes make up the majority (Zhang et al., 2021).

Various methods have been developed for LTR, such as class-balanced loss (CB) (Cui et al., 2019), augmenting samples of tail classes (Wang et al., 2021), two-stage learning (Kang et al., 2020), and enhancing feature extractors (Liu et al., 2023; Yang et al., 2022); see Appendix A.2 for more related research. Alshammari et al. (2022) proposed a simple method, called *weight balancing (WB)*, that empirically outperforms previous complex state-of-the-art methods. WB simply combines two classic techniques, weight decay (WD) (Hanson & Pratt, 1989) and MaxNorm (Hinton et al., 2012), with two-stage learning. WD and MaxNorm is known to prevent overlearning (Hinton, 1989); however, it is not known why WB significantly improves the LTR performance.

**Contribution** In this work, we analyze the effectiveness of WB in LTR focusing on neural collapse (NC) (Papyan et al., 2020) and the cone effect (Liang et al., 2022). We first decompose WB into five components: WD, MaxNorm, cross entropy (CE), CB, and two-stage learning. We then show that each component has the following useful properties.

- 1st stage: WD and CE increase the Fisher's discriminant ratio (FDR) (Fisher, 1936) of features.
    - Degrade the inter-class cosine similarities (Theorem 1 in Sec. 4.2).
    - Decrease the scaling parameters of batch normalization (BN) (Sec. 4.3). This has a positive effect on feature training.
    - Facilitate improvement of FDR as features pass through layers (Sec. 4.4).
- 1st stage: WD increases the norms of features for tail classes (Sec. 4.5).
- 2nd stage: WD and CB perform implicit logit adjustment (LA) by making the norm of classifier's weights higher for tail classes. MaxNorm facilitates this effect. This stage does not work well for datasets with a small class number (Theorem 2 in Sec. 5.1).

The above analysis is useful in the following points: 1. it provides a guideline for the design of learning in LTR; 2. our theorem offers an insight into how to suppress the cone effect (Liang et al., 2022) that negatively affects deep learning classification models; 3. WB can be further simplified by removing the second stage. Specifically, we only need to learn in LTR by WD, feature regularization (FR), and an equiangular tight frame (ETF) classifier for the linear layer to extract features that are more linearly separable and adjusting the norm of classifier's weights by LA after the training.

## 2 RELATED WORK

Papyan et al. (2020) investigated the set of phenomena that occurs when the terminal phase of training, which they termed NC. NC can be briefly described as follows. Feature vectors converge to their class means (NC1); the class means of feature vectors converge to a simplex ETF (Strohmer & Heath, 2003) (NC2); the class means of feature vectors and the corresponding weights of the linear classifier converge in the same direction (NC3); and when making predictions, models converge to predict the class, the mean feature vector of which is closest to the feature in Euclidean distance (NC4).

Papyan et al. (2020) claimed that NC leads to increased generalization accuracy and robustness to adversarial samples. The number of studies have been conducted on the conditions under which NC occurs (Han et al., 2022; Ji et al., 2022; Lu & Steinerberger, 2021; Papyan et al., 2020). Rangamani & Banburski-Fahey (2022) theoretically and empirically showed that the occurrence of NC needs WD.

There have also been studied on NC when the model is trained with imbalanced data. Fang et al. (2021) theoretically and experimentally proved "minority collapse" in which features corresponding to classes with small numbers of samples tend to converge in the same direction, even if they belong to different classes. Yang et al. (2022) attempted to solve this problem by fixing the weights of the linear layers to an ETF. Thrampoulidis et al. (2022) tried to generalize NC to imbalanced data by demonstrating that the features converge towards simplex-encoded-labels interpolation (SELI), an extension of ETF in the terminal phase of training. They also demonstrated that minority collapse does not occur under the appropriate regularization including WD.

WD is a regularization method often used for deep neural networks (DNNs) with BN (Ioffe & Szegedy, 2015), which is scale invariant. Zhang et al. (2019) revealed that WD increases the effective learning rate which facilitates regularization in such models. Training of networks containing BN layers with WD is often studied in terms of training dynamics (Li & Arora, 2020; Lobacheva et al., 2021; Wan et al., 2021). Summers & Dinneen (2020) and Kim et al. (2022) studied whether to apply WD to the scaling and shifting parameters of the BN for such networks. WD is also known to have many other positive effects on DNNs, e.g., smoothing loss landscape (Li et al., 2018; Lyu et al., 2022), causing NC (Rangamani & Banburski-Fahey, 2022), and making filters sparser (Mehta et al., 2019) and low-ranked (Galanti et al., 2023). WD is also effective for imbalanced data (Alshammari et al., 2022; Thrampoulidis et al., 2022).

Liang et al. (2022) found a phenomenon they termed "cone effect" observed in neural networks. The phenomenon is a tendency in which features from DNNs with activation functions are prone to have high cosine similarities. It is widely observed regardless of the modality of data, the structure of models, and whether the models are trained. It also indicates that features from different classes tend to exhibit high cosine similarity. Thus, it has a negative impact on DNNs for classification since features with lower inter-class cosine similarity are more linearly separated. We also reconfirmed such phenomena in Sec. 4.2. Our analysis provides guidelines to prevent the effect.

## 3  PRELIMINARIES

This section defines the notations used in the paper and presents the strategies for LTR. See Appendix A.1 for the table of these notations. Suppose a multiclass classification with $C$ classes by samples $\mathcal{X} \subset \mathbb{R}^p$ and labels $\mathcal{Y} \equiv \{1, 2, \ldots, C\}$. The training dataset $\mathcal{D} \equiv \{(\mathbf{x}_i, y_i) | \mathbf{x}_i \in \mathcal{X}, y_i \in \mathcal{Y}\}_{i=1}^{N}$ consists of $\mathcal{D}_k \equiv \{(\mathbf{x}_i, y_i) \in \mathcal{D} \mid y_i = k\}$. We define $N$ as $\|\mathcal{D}\|$ and $N_k$ as $\|\mathcal{D}_k\|$, the number of samples. Without loss of generality, assume the classes are sorted in descending order by the number of samples. In other words, $\forall k \in \{1, 2, \ldots, C-1\}$, $N_k \geq N_{k+1}$ holds. Imbalance factor $\rho = \frac{N_1}{N_C} = \frac{\max_k N_k}{\min_k N_k}$ indicates the extent to which the training dataset is imbalanced and $\rho \gg 1$ holds in LTR. Therefore, the number of samples in each class $N_k$ satisfies $N_k = N_1 \rho^{-\frac{k-1}{C-1}}$. Define $\overline{N}$ as $C\left(\sum_{k=1}^{C}\left(\frac{1}{N_k}\right)\right)^{-1}$, the harmonic mean of the number of samples per class. In LTR, the test dataset $\mathcal{D}'$ used for accuracy evaluation is class-balanced, i.e., each class has the same number of samples.

Consider a network $\boldsymbol{f}(\cdot; \boldsymbol{\Theta}) : \mathcal{X} \to \mathbb{R}^C$ parameterized by $\boldsymbol{\Theta} = \{\boldsymbol{\theta}_l\}$. It outputs a logit $\mathbf{z}_i = \boldsymbol{f}(\mathbf{x}_i; \boldsymbol{\Theta})$. The network is further divided into a feature extractor $\boldsymbol{g}(\cdot; \boldsymbol{\Theta}_g) : \mathcal{X} \to \mathbb{R}^d$ and a classifier $\boldsymbol{h}(\cdot; \boldsymbol{\Theta}_h) : \mathbb{R}^d \to \mathbb{R}^C$ with $d$ as the number of dimensions for features. This means $\boldsymbol{f}(\mathbf{x}_i; \boldsymbol{\Theta}) = \boldsymbol{h}(\boldsymbol{g}(\mathbf{x}_i; \boldsymbol{\Theta}_g); \boldsymbol{\Theta}_h)$. We often abbreviate $\boldsymbol{g}(\mathbf{x}_i; \boldsymbol{\Theta}_g)$ to $\boldsymbol{g}(\mathbf{x}_i)$. Let $\boldsymbol{\mu}_k \equiv \frac{1}{N_k} \sum_{(\mathbf{x}_i, k) \in \mathcal{D}_k} \boldsymbol{g}(\mathbf{x}_i)$ be the inner-class mean of the features for class $k$ and $\boldsymbol{\mu} \equiv \frac{1}{C} \sum_{k=1}^{C} \boldsymbol{\mu}_k$ be the mean of the inner-class mean of the features. We used a linear layer for the classifier. In other words, $\boldsymbol{h}(\mathbf{v}; \boldsymbol{\Theta}_h) = \mathbf{W}^\top \mathbf{v}$ holds for a feature $\mathbf{v} \in \mathbb{R}^d$ with $\mathbf{W} = \mathbb{R}^{d \times C}$ as a weight matrix. Denote the $k$th column vector of $\mathbf{W}$ by $\mathbf{w}_k$; thus, $\boldsymbol{\Theta}_h = \{\mathbf{w}_k\}$. The loss function is denoted by $\ell(\mathbf{z}_i, y_i) : \mathbb{R}^C \times \mathbb{R}^C \to \mathbb{R}$. Let $\ell_{\text{CE}}$ and $\ell_{\text{CB}}$ be the loss function of CE and CB, respectively. By rewriting $F(\boldsymbol{\Theta}; \mathcal{D}) \equiv \frac{1}{N} \sum_{i=1}^{N} \ell(\boldsymbol{f}(\mathbf{x}_i; \boldsymbol{\Theta}), y_i)$, parameters $\boldsymbol{\Theta}$ are optimized as follows in the absence of regularization:

$$\boldsymbol{\Theta}^* = \arg\min_{\boldsymbol{\Theta}} F(\boldsymbol{\Theta}; \mathcal{D}). \tag{1}$$

We evaluated the method by accuracy on test dataset and FDR. FDR is the ratio of the inter-class variance to the inner-class variance and indicates the ease of linear separation of the features. For example, the FDR of training features is $\text{Tr}(S_W^{-1} S_B)$ where $S_B = \sum_{k=1}^{C} N_k(\boldsymbol{\mu}_k - \boldsymbol{\mu})(\boldsymbol{\mu}_k - \boldsymbol{\mu})^\top$ and $S_W = \sum_{k=1}^{C} \sum_{\mathbf{x}_i \in \mathcal{D}_k} (\mathbf{x}_i - \boldsymbol{\mu}_k)(\mathbf{x}_i - \boldsymbol{\mu}_k)^\top$. The FDR of test features is similar.

**Regularization Methods**  We implemented WD as L2 regularization with a hyperparameter $\lambda$ because we used stochastic gradient descent (SGD) for the optimizer. Thus, optimization is written as $\boldsymbol{\Theta}^* = \arg\min_{\boldsymbol{\Theta}} F(\boldsymbol{\Theta}; \mathcal{D}) + \frac{\lambda}{2} \sum_{k \in \mathcal{Y}} \|\mathbf{w}_k\|_2^2$. MaxNorm is a regularization that places a restriction on the upper bound of the norm of weights. As with Alshammari et al. (2022), for the convenience of hyperparameter tuning, the weights to be constrained are only the those belonging to the classifier $\boldsymbol{\Theta}_h \subset \boldsymbol{\Theta}$. Thus, we compute $\boldsymbol{\Theta}^* = \arg\min_{\boldsymbol{\Theta}} F(\boldsymbol{\Theta}; \mathcal{D})$, s.t. $\forall\, \mathbf{w}_k \in \boldsymbol{\Theta}_h$, $\|\mathbf{w}_k\|_2^2 \leq \eta_k^2$, where $\eta_k$ is a hyperparameter. Since constrained optimization is difficult to solve in neural networks, we implemented it using projected gradient descent. In our implementation, this projects the weights that do not satisfy the constraint to the range where the constraint is satisfied by updating $\mathbf{w}_k$ to $\min\left(1, \frac{\eta_k}{\|\mathbf{w}_k\|_2}\right) \mathbf{w}_k$.

**Weight Balancing**  WB adopts two-stage training (Kang et al., 2020). In the first stage, the parameters of the entire model $\boldsymbol{\Theta}$ are optimized with CE using WD. In the second stage, $\boldsymbol{\Theta}_g$ is fixed and only $\boldsymbol{\Theta}_h$ is optimized with CB using WD and MaxNorm. For simplicity, let $\beta$ of CB be 1 in this paper. Thus, $\ell_{\text{CB}}(\mathbf{z}_i, y_i)$ is $-\frac{\overline{N}}{N_{y_i}} \log\left(\frac{\exp((\mathbf{z}_i)_{y_i})}{\sum_{j=1}^{C} \exp((\mathbf{z}_i)_j)}\right)$.[1] Therefore, in the second stage of WB, $F_{\text{WB}}(\mathbf{W}; \mathcal{D}) \equiv \frac{1}{N} \sum_{i=1}^{N} \ell_{\text{CB}}(\mathbf{z}_i, y_i) + \frac{\lambda}{2} \sum_{k \in \mathcal{Y}} \|\mathbf{w}_k\|_2^2$ is optimized for $\mathbf{W}$. Note that we do not consider MaxNorm in this paper, as we reveal that it only changes the initial values and do not impose any intrinsic constraints on the optimization presented in Appendix. A.5.

**Logit Adjustment**  Multiplicative LA (Kim & Kim, 2020) changes the norm of the per-class weights of the linear layer. The weights of the linear layer corresponding to class $k$, $\mathbf{w}_k'$ are adjusted

---

[1]In the paper of Cui et al. (2019), $\overline{N}$ is set to be 1, but in their implementation, $\overline{N}$ is equal to the harmonic mean. We adopt the latter since our experiments were based on their implementation.

Table 1: FDRs for each dataset of models trained with each method. A higher FDR indicates that features are more easily linearly separable. For all datasets, the method with WD or CE produces a higher FDR and using both results in the highest FDR.

| Method | CIFAR10-LT | | CIFAR100-LT | | mini-ImageNet-LT | |
| --- | --- | --- | --- | --- | --- | --- |
| | Train | Test | Train | Test | Train | Test |
| CE w/o WD | $8.17 \times 10^1$ | $2.17 \times 10^1$ | $1.28 \times 10^2$ | $4.16 \times 10^1$ | $7.56 \times 10^1$ | $4.28 \times 10^1$ |
| CB w/o WD | $4.43 \times 10^1$ | $1.50 \times 10^1$ | $8.17 \times 10^1$ | $2.42 \times 10^1$ | $4.68 \times 10^1$ | $2.93 \times 10^1$ |
| CE w/ WD | $\mathbf{2.60 \times 10^3}$ | $\mathbf{3.89 \times 10^1}$ | $\mathbf{2.87 \times 10^4}$ | $\mathbf{1.07 \times 10^2}$ | $\mathbf{6.58 \times 10^2}$ | $\mathbf{1.01 \times 10^2}$ |
| CB w/ WD | $3.39 \times 10^2$ | $3.04 \times 10^1$ | $2.12 \times 10^4$ | $6.74 \times 10^1$ | $4.84 \times 10^2$ | $6.84 \times 10^1$ |
| WD w/o BN | $2.19 \times 10^2$ | $3.42 \times 10^1$ | $5.77 \times 10^2$ | $7.95 \times 10^1$ | $1.61 \times 10^2$ | $6.73 \times 10^1$ |
| WD fixed BN | $\mathbf{1.63 \times 10^3}$ | $\mathbf{4.16 \times 10^1}$ | $\mathbf{2.04 \times 10^4}$ | $\mathbf{1.05 \times 10^2}$ | $\mathbf{3.94 \times 10^2}$ | $\mathbf{1.04 \times 10^2}$ |

to $\frac{1}{\mathbb{P}(Y=k)^\gamma} \frac{\mathbf{w}_k}{\|\mathbf{w}_k\|_2}$, where $\gamma > 0$ is a hyperparameter. Kim & Kim (2020) trained using projected gradient descent so that the weights of the linear layer always satisfy $\forall k, \|\mathbf{w}_k\|_2 = 1$, but in our study, for comparison, we trained with regular gradient descent, normalized the norm post-hoc, and then adjusted the norm. Additive LA (Menon et al., 2020) is a method to adjust the output to minimize the average per-class error by adding a different constant for each class to the logit at prediction. That is, the logit $\mathbf{z}_i$ at prediction is adjusted to $\mathbf{z}'_i$ as $(\mathbf{z}'_i)_k = (\mathbf{z}_i)_k - \tau \log \mathbb{P}(Y = k)$, where $\tau$ is a hyperparameter. LA generally increases the probability of classifying inputs into tail classes.

**ETF Classifier**   NC indicates that the matrix of classifier weights in deep learning models converges to an ETF when trained with CE (Papyan et al., 2020). The idea of ETF classifiers is to train only the feature extractor by fixing the linear layer to an ETF from the initial step. ETF classifiers' weights $\mathbf{W} \in \mathbb{R}^{d \times C}$ satisfy $\mathbf{W} = \sqrt{E_W \frac{C}{C-1}} \mathbf{U} \left( \mathbf{I}_C - \frac{1}{C} \mathbf{1}_C \mathbf{1}_C^\top \right)$, where $\mathbf{U} \in \mathbb{R}^{d \times C}$ is a matrix such that $\mathbf{U}^\top \mathbf{U}$ is an identity matrix, $\mathbf{I}_C \in \mathbb{R}^{C \times C}$ is an identity matrix, and $\mathbf{1}_C$ is a $C$-dimensional vector, all elements in which are 1. Also, $E_W$ is a hyperparameter, which was set to 1 in our experiments.

## 4   ROLE OF FIRST TRAINING STAGE IN WB

First, we analyze the effect of WD and CE in the first training stage. In Sec. 4.1, we present the datasets and models used in the analysis. Then, we examine the cosine similarity and FDR of the features from the models trained with each training method for the first stage of training in Secs. 4.2, 4.3, and 4.4. In Sec. 4.5, we identify the properties of training features trained with WD, which is the key to the success of WB. The results of the experiments that is not included in the main text are presented in Appendix A.5.

### 4.1   SETTINGS

We used CIFAR10, CIFAR100 (Krizhevsky, 2009), mini-ImageNet (Vinyals et al., 2016), and ImageNet (Deng et al., 2009) as the datasets and followed Cui et al. (2019), Vigneswaran et al. (2021), and Liu et al. (2019) to create long-tailed datasets, CIFAR10-LT, CIFAR100-LT, mini-ImageNet-LT, and ImageNet-LT. We also experimented with a tabular dataset, Helena (Guyon et al., 2019), to verify the effectiveness of our method for other than image dataset. Each class is classified into one of three groups, *Many*, *Medium*, and *Few*, depending on the number of training samples $N_k$.

We used ResNeXt50 (Xie et al., 2017) for ImageNet-LT and ResNet34 (He et al., 2016) for the other datasets. In WB, parameters were trained in two stages (Kang et al., 2020); in the first stage, we trained the weights of the entire model $\Theta$, while in the second stage, we trained the weights of the classifiers $\Theta_h$ with the weights of the feature extractors $\Theta_g$ fixed.

To investigate the properties of neural network models, we also experimented with models using a multilayer perceptron (MLP) and a residual block (ResBlock) (He et al., 2016). The number of blocks is added after the model name to distinguish them (e.g., MLP3). We trained and evaluated these models using MNIST (Lecun et al., 1998).

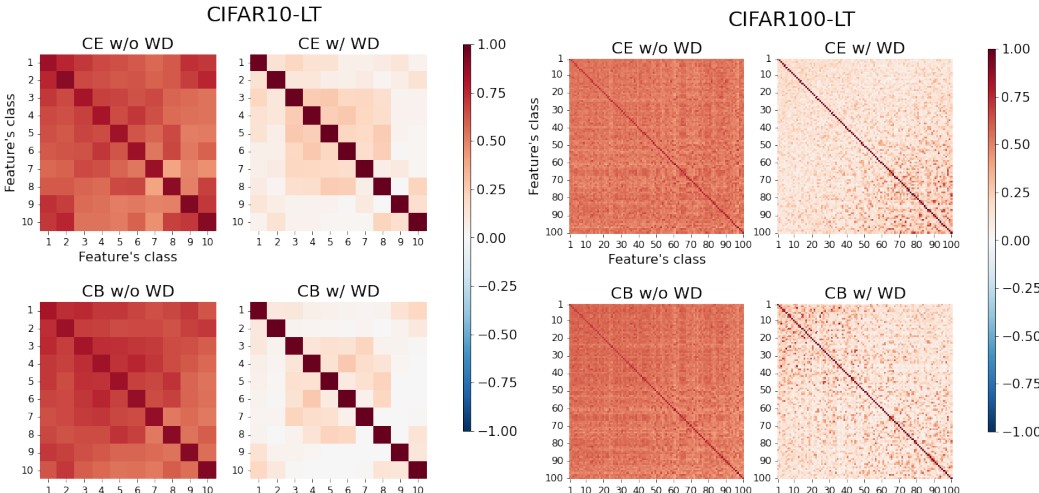

Figure 1: Heatmaps showing average cosine similarities of training features between two classes. WD maintains high cosine similarity between the same classes and reduces the cosine similarity between the different classes.

For training losses, we used CE unless otherwise stated. See Appendix A.4 for detailed settings.

## 4.2 WD AND CE DEGRADE INTER-CLASS COSINE SIMILARITIES

We first examined the role of WD in the first stage of training. We trained the model in two ways: without WD (w/o WD) and with WD (w/ WD). In addition, we compared two types of loss: one using CE and the other using CB, for a total of four training methods. We investigated the FDR, the inter-class and inner-class mean cosine similarity of the features, and the norm of the mean features per class. Due to space limitations, some of results of the mini-ImageNet-LT excluding FDR are presented in Appendix A.5.

Table 1 lists the FDRs of the models trained with each method, and Figure 1 shows the cosine similarities of the training features per class. It can be seen that the combination of WD and CE achieves the highest FDR. This fact is consistent with the conclusion that WD is necessary for the NC in balanced datasets, as shown by Rangamani & Banburski-Fahey (2022) and suggests that this may also hold for long-tailed data.

When WD is not applied, the cosine similarity is generally higher even between features of different classes. This is natural because neural networks have the cone effect found in Liang et al. (2022). The cone effect is a phenomenon in which features from a DNN tend to have high cosine similarities to each other even if they belong to different classes. However, the methods with WD result in lower cosine similarities between features of different classes and higher cosine similarities between features of the same classes, leading to a higher FDR. This shows that WD prevents the cone effect, as supported by the following theorem.

**Theorem 1.** *For all $(\mathbf{x}_i, y_i), (\mathbf{x}_j, y_j) \in \mathcal{D}$ s.t. $y_i \neq y_j$, if $\mathbf{W}$ is an ETF and there exists $\epsilon$ and $L$ s.t. $\left\| \frac{\partial \ell_{\mathrm{CE}}}{\partial \boldsymbol{g}(\mathbf{x}_i)} \right\|_2, \left\| \frac{\partial \ell_{\mathrm{CE}}}{\partial \boldsymbol{g}(\mathbf{x}_j)} \right\|_2 \leq \epsilon < \frac{1}{C}$ and $\|\boldsymbol{g}(\mathbf{x}_i)\|_2, \|\boldsymbol{g}(\mathbf{x}_j)\|_2 \leq L \leq 2\sqrt{2} \log(C-1)$, the following holds:*

$$\cos\left(\boldsymbol{g}(\mathbf{x}_i), \boldsymbol{g}(\mathbf{x}_j)\right) \leq 2\delta\sqrt{1-\delta^2}, \tag{2}$$

*where $\delta \equiv \frac{1}{L} \frac{C-1}{C} \log\left(\frac{(C-1)(1-\epsilon)}{\epsilon}\right) \in \left(\frac{1}{\sqrt{2}}, 1\right]$ and $\cos(\cdot, \cdot)$ means cosine similarity of the two vectors.*

This theorem states that the cone effect is suppressed when the following two conditions hold: 1. the weight matrix of the linear layer is an ETF; 2. the norms of the features are sufficiently small. Sufficient training causes NC and makes the weight matrix of the linear layer converge to an ETF

Table 2: Means and standard deviations of all scaling parmeters in BN layers of models trained with each method for each dataset. WD greatly reduces the average scaling parameter.

| Method | CIFAR10-LT | CIFAR100-LT | mini-ImageNet-LT |
|---|---|---|---|
| CE w/o WD | $1.00 \pm 0.0218$ | $1.00 \pm 0.0289$ | $0.998 \pm 0.0498$ |
| CE w/ WD | $0.0395 \pm 0.0273$ | $0.0649 \pm 0.0548$ | $0.0863 \pm 0.0707$ |

Table 3: Means and standard deviations of all shifting parmeters in BN layers of models trained with each method for each dataset.

| Method | CIFAR10-LT | CIFAR100-LT | mini-ImageNet-LT |
|---|---|---|---|
| CE w/o WD | $-0.0236 \pm 0.0255$ | $-0.0444 \pm 0.0348$ | $-0.0904 \pm 0.0698$ |
| CE w/ WD | $-0.00310 \pm 0.0244$ | $-0.00667 \pm 0.0309$ | $-0.0186 \pm 0.0448$ |

(Papyan et al., 2020). ETF Classifiers are also effective: thery are methods to fix the weight matrix of the linear layer to an ETF from the beginning to satisfy this condition. As for the second condition, WD contributes to satisfy the condition. In fact, as shown in Figure 2, WD also indirectly degrades the norm of the features by regularizing the weights causing low inter-class cosine similarity. This also suggests that explicit FR leads to better feature extractor training. We consider that in Sec. 5.2.

### 4.3 WD AND CE DECREASE SCALING PARAMETERS OF BN

Next, we split the effect of WD on the model into that on the convolution layers and that on the BN layers. To examine the impact on convolution, we analyzed the FDR of the features from the models trained with WD applied only to the convolution layers (WD w/o BN). The bottom halves of Table 1 shows that the FDRs of the methods with restricted WD application. These results indicate that when WD is applied only to the convolution layers, the FDR improves sufficiently, which means that enabling WD for the convolution layers is essential for improving accuracy. One reason for this may be an increase in the effective learning rate (Zhang et al., 2019).

**Small scaling parameters of BN facilitates feature learning** We also examined the effect on BN layers and found that WD reduces the mean of the BN's scaling parameters; see Tables 2 and 3. Note that the standard deviation of the shifting parameters remains almost unchanged. We investigated how the FDR changes when models are trained with the scaling parameters of BN fixed to one common small value (WD fixed BN). In this method, we applied WD only to the convolution layers and fixed the shifting parameters of BN to 0. We selected the optimal value for the scaling parameters from $\{0.01, 0.02, \ldots, 0.20\}$ using the validation dataset. Table 1 shows that a small value of the scaling parameters of BN is crucial for boosting FDR; even setting the same value for all scaling parameters in the entire model also works. In this case, the scaling parameters of BN does not directly affect FDR, as it only multiplies the features uniformly by a constant. This suggests that the improvement in FDR caused by applying WD to BN layers is mainly because smaller scaling parameters have a positive effect on the learning dynamics and improve FDR. Although Kim et al. (2022) attribute this to the increase in the effective learning rate of the scaling parameters, they have a positive effect on FDR even when the scaling parameters are not trained, suggesting that there are other significant effects. For example, smaller scaling parameters reduce the norm of the feature, promoting the effects discussed in Sec. 4.2. We also examined the effect of high relative variance in shifting parameters of BN in Appeindix A.5.

### 4.4 WD AND CE FACILITATE IMPROVEMENT OF FDR AS FEATURES PASS THROUGH LAYERS

Table 4 compares the FDR of two sets of features; ones from each model trained with each method (before) and ones output from the model, passed through a randomly initialized linear layer, then ReLU-applied (after). This experiment revealed that the features obtained from the models trained with WD have a bias to improve their FDR regardless of the class imbalance. The detailed experimental set-up is described in Appendix A.5. While training without WD does not increase the FDR in

Table 4: FDRs of features from each model trained with each method and features passed through randomly initialized linear layer and ReLU after model. C10, C100, and mIm are the abbreviations for CIFAR10, CIFAR100, and mini-ImageNet respectively. Red text indicates higher values than before, blue text indicates lower values. The FDR of features obtained from models trained with WD improves by passing the features through a random initialized layer and ReLU.

| Model | Dataset | CE | | WD w/o BN | | WD | |
|---|---|---|---|---|---|---|---|
| | | before | after | before | after | before | after |
| ResNet34 | C100 | $7.51 \times 10^1$ | $7.11 \times 10^1$ | $2.29 \times 10^2$ | $2.34 \times 10^2$ | $3.98 \times 10^2$ | $4.10 \times 10^2$ |
| | C100-LT | $4.16 \times 10^1$ | $4.02 \times 10^1$ | $7.95 \times 10^1$ | $7.89 \times 10^1$ | $1.07 \times 10^2$ | $1.13 \times 10^2$ |
| | C10 | $4.77 \times 10^1$ | $5.56 \times 10^1$ | $7.96 \times 10^1$ | $1.02 \times 10^2$ | $1.81 \times 10^2$ | $2.35 \times 10^2$ |
| | C10-LT | $2.17 \times 10^1$ | $2.36 \times 10^1$ | $3.42 \times 10^1$ | $3.94 \times 10^1$ | $3.89 \times 10^1$ | $4.30 \times 10^1$ |
| | mIm | $7.34 \times 10^1$ | $6.66 \times 10^1$ | $1.81 \times 10^2$ | $1.86 \times 10^2$ | $3.63 \times 10^2$ | $3.93 \times 10^2$ |
| | mIm-LT | $4.28 \times 10^1$ | $3.97 \times 10^1$ | $6.73 \times 10^1$ | $6.30 \times 10^1$ | $1.01 \times 10^2$ | $1.08 \times 10^2$ |
| MLP3 | MNIST | $1.58 \times 10^2$ | $1.54 \times 10^2$ | $2.48 \times 10^2$ | $3.53 \times 10^2$ | $2.36 \times 10^2$ | $7.96 \times 10^2$ |
| MLP4 | | $1.98 \times 10^2$ | $1.96 \times 10^2$ | $3.60 \times 10^2$ | $4.97 \times 10^2$ | $4.12 \times 10^2$ | $1.43 \times 10^3$ |
| MLP5 | | $2.44 \times 10^2$ | $2.37 \times 10^2$ | $4.91 \times 10^2$ | $6.76 \times 10^2$ | $6.10 \times 10^2$ | $2.47 \times 10^3$ |
| ResBlock1 | | $1.39 \times 10^2$ | $1.36 \times 10^2$ | $2.20 \times 10^2$ | $2.69 \times 10^2$ | $2.44 \times 10^2$ | $5.81 \times 10^2$ |
| ResBlock2 | | $1.66 \times 10^2$ | $1.59 \times 10^2$ | $2.97 \times 10^2$ | $3.65 \times 10^2$ | $4.44 \times 10^2$ | $9.55 \times 10^2$ |

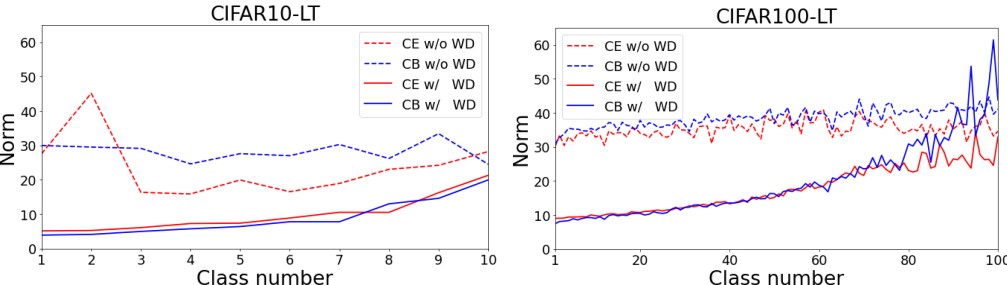

Figure 2: Norm of mean per-class training features produced from models trained with each method. Features learned with methods with WD all demonstrate that the norms of the *Many* classes' features tend to be smaller than those of the *Few* classes.

most cases, the features from models with WD-applied convolution improve the FDR over the before FDR in most cases. Moreover, the after-FDR increases significantly when WD is applied to the whole model. Note that this phenomenon is valid when the features go through only one randomized layer. Experiments have also shown that passing features through multiple layers of randomly initialized linear layers and ReLUs does not necessarily improve FDR; see Appendix A.5. These results mean that only increasing randomized layers does not enhance FDR. However, features trained with WD are less likely to be degraded even when going through one randomized layer. This suggests that WD and CE facilitate the training of models.

### 4.5    WD INCREASES NORMS OF FEATURES FOR TAIL CLASSES

We also investigated the properties appearing in the norm of the training features from each model investigated in Sec. 4.2; Sec. 5.1 reveals that this property is crucial to the effectiveness of WB. Figure 2 shows the norm of the per-class mean training features. Note that this phenomenon does not occur with test features. The method without WD shows almost no relationship between the number of samples and the norm of the features for each class. However, when WD is applied, the norms of the *Many*-class features drop significantly, to between one-half and one-fifth that of *Few* classes. While this phenomenon has been observed in step-imbalanced data (Thrampoulidis et al., 2022), we found that it also occurs in long-tailed data.

## 5   ROLE OF SECOND TRAINING STAGE IN WB

In this section, we theoretically analyze how the second stage of training operates in Sec. 5.1, which leads to further simplification and empirical performance improvement of WB in Sec. 5.2.

### 5.1   WD AND CB PERFORM IMPLICIT LOGIT ADJUSTMENT

We found that WB encourages the model to output features with a high FDR in the first stage of training. How does the second stage of training improve accuracy? Alshammari et al. (2022) found that WB trains models so that the norm of the linear layer becomes more significant for the tail classes in the second stage of training but does not explicitly train in this way. We found the following theorem that shows the second stage of WB is equivalent to multiplicative LA under certain assumptions. In the following theorem, we assume $\boldsymbol{\mu} = \mathbf{0}$, which is valid when NC occurs, i.e., when per-class mean features follow ETF or SELI (Papyan et al., 2020; Thrampoulidis et al., 2022). See Appendix. A.3 for the theorem and proof when this does not hold.

**Theorem 2.** *Assume* $\boldsymbol{\mu} = \mathbf{0}$*. For any* $k \in \mathcal{Y}$*, if there exists* $\mathbf{w}_k^*$ *s.t.* $\left. \frac{\partial F_{\mathrm{WB}}}{\partial \mathbf{w}_k} \right|_{\mathbf{w}_k = \mathbf{w}_k^*} = \mathbf{0}$ *and* $\|\mathbf{w}_k^*\|_2 = O\left(\frac{1}{\lambda \rho C}\right)$*, we have*

$$\forall k \in \mathcal{Y}, \left\| \mathbf{w}_k^* - \frac{\overline{N}}{\lambda N} \boldsymbol{\mu}_k \right\|_2 = O\left(\frac{1}{\lambda^2 \rho^2 C^2}\right), \tag{3}$$

where $O$ is a Bachmann-Landau O-notation. For example, the notation in the form $O\left(\frac{1}{x}\right)$ indicates that the term is at most a constant multiple of $\frac{1}{x}$ if $x$ is sufficiently large. Note that $\frac{\overline{N}}{\lambda N}$ is independent of $k$. This theorem states that if the number of classes or the imbalance factor is sufficiently large, and if NC has occurred in the first stage, there exists linear layer weights that are constant multiples of the corresponding features and sufficiently close to the stationary point in the optimization. In other words, if the previous conditions are satisfied, the norm of the linear layer is proportional to the norm of the corresponding per-class mean feature. We show that the norm of training features are larger for *Few* classes in Sec. 4.5. Thus, the linear layer is trained to be larger for the *Few* classes. Besides, the features and the corresponding linear layers' weights are aligned when NC occurs, so the orientation of the linear layers' weights hardly changes during this training. This is the same operation as when multiplicative LA makes adjustments to long-tailed data, increasing the norm of the linear layer for the tail class (Kim & Kim, 2020). Indeed, if there exists some $c_0$ and $\gamma_0$, and for all $k$, $\|\boldsymbol{\mu}_k\|_2 = c_0 \mathbb{P}(Y = k)^{\gamma_0}$ holds, then the second-stage training is equivalent to multiplicative LA. In this theorem, we do not consider MaxNorm because we found that MaxNorm makes the norm of the weights in the linear layer closer to the same value before training, only facilitating convergence. For more details, see Appendix A.5.

Note also that this theorem does not guarantee that the second stage of WB is an operation equivalent to multiplicative LA when the number of classes is small. In fact, even though the norm of the training features is larger for *Few* classes (Figure 2), WB fails to make the norm of the weights greater for *Few* classes in CIFAR10-LT: see Appendix A.5. This suggests that replacing the second stage of training with LA would be more generic and we confirm its validity in the following section.

### 5.2   IS TWO-STAGE LEARNING REALLY NECESSARY?

We experimentally demonstrated the role of each training stage of WB as indicated by Theorems 1 and 2. For this purpose, we develop a method with a combination of essential operations and verify its high accuracy and FDR. In the first stage of training, FR and an ETF classifier are used to satisfy the conditions of Theorem 1. We fix the linear layer and train the feature extractor with CE, WD, and FR; then adjust the norm of the linear layer with multiplicative LA. To show that these are effective operations implicitly carried out with WB, we compare the combination with WB. For the ablation study, we experimented by comparing WB with the following methods.

- Training with CE or CB only (CE, CB).
- Training with WD and CE (WD).

Table 5: FDRs and accuracy of models trained with each method for CIFAR100-LT. In addition to WD, training with FR and an ETF classifier further improves the FDR. However, only using these does not greatly improve accuracy; LA increases the accuracy of *Medium* and *Few* classes and enhances the average accuracy significantly as a result. In all cases, multiplicative LA improves the accuracy more than additive LA.

| Method | LA | FDR | | Accuracy (%) | | | |
| --- | --- | --- | --- | --- | --- | --- | --- |
| | | Train | Test | *Many* | *Medium* | *Few* | Average |
| CE | N/A | $1.28 \times 10^2_{\pm 0.9 \times 10^1}$ | $4.16 \times 10^1_{\pm 1.0 \times 10^0}$ | $64.6_{\pm 0.9}$ | $36.9_{\pm 0.8}$ | $11.9_{\pm 0.7}$ | $38.5_{\pm 0.6}$ |
| CB | N/A | $8.17 \times 10^1_{\pm 8.0 \times 10^0}$ | $2.42 \times 10^1_{\pm 0.6 \times 10^0}$ | $47.6_{\pm 1.0}$ | $23.2_{\pm 0.6}$ | $5.61_{\pm 0.4}$ | $26.1_{\pm 0.4}$ |
| WD | N/A | $2.87 \times 10^4_{\pm 6.0 \times 10^3}$ | $1.07 \times 10^2_{\pm 0.2 \times 10^1}$ | $75.9_{\pm 0.5}$ | $45.3_{\pm 0.6}$ | $13.9_{\pm 0.7}$ | $46.0_{\pm 0.4}$ |
| | Add | $2.87 \times 10^4_{\pm 6.0 \times 10^3}$ | $1.07 \times 10^2_{\pm 0.2 \times 10^1}$ | $70.7_{\pm 0.9}$ | $45.7_{\pm 0.9}$ | $\mathbf{30.9_{\pm 0.9}}$ | $49.6_{\pm 0.4}$ |
| | Mult | $2.87 \times 10^4_{\pm 6.0 \times 10^3}$ | $1.07 \times 10^2_{\pm 0.2 \times 10^1}$ | $72.6_{\pm 0.7}$ | $48.5_{\pm 0.8}$ | $29.5_{\pm 0.9}$ | $50.8_{\pm 0.4}$ |
| WB | N/A | $2.94 \times 10^4_{\pm 6.0 \times 10^3}$ | $1.07 \times 10^2_{\pm 0.2 \times 10^1}$ | $73.8_{\pm 0.7}$ | $50.2_{\pm 0.6}$ | $25.6_{\pm 1.0}$ | $50.6_{\pm 0.2}$ |
| WD&ETF | N/A | $3.33 \times 10^4_{\pm 2.0 \times 10^3}$ | $1.13 \times 10^2_{\pm 0.1 \times 10^1}$ | $76.3_{\pm 0.3}$ | $46.0_{\pm 0.4}$ | $15.5_{\pm 0.6}$ | $46.9_{\pm 0.2}$ |
| | Add | $3.33 \times 10^4_{\pm 2.0 \times 10^3}$ | $1.13 \times 10^2_{\pm 0.1 \times 10^1}$ | $73.8_{\pm 0.4}$ | $48.9_{\pm 0.3}$ | $25.8_{\pm 0.4}$ | $50.2_{\pm 0.2}$ |
| | Mult | $3.33 \times 10^4_{\pm 2.0 \times 10^3}$ | $1.13 \times 10^2_{\pm 0.1 \times 10^1}$ | $70.4_{\pm 0.7}$ | $51.4_{\pm 0.3}$ | $\mathbf{31.7_{\pm 0.6}}$ | $51.7_{\pm 0.3}$ |
| WD&FR &ETF | N/A | $\mathbf{8.81 \times 10^4_{\pm 2.0 \times 10^3}}$ | $\mathbf{1.22 \times 10^2_{\pm 0.1 \times 10^1}}$ | $\mathbf{77.9_{\pm 0.3}}$ | $46.8_{\pm 1.0}$ | $15.3_{\pm 0.3}$ | $47.6_{\pm 0.5}$ |
| | Add | $\mathbf{8.85 \times 10^4_{\pm 2.0 \times 10^3}}$ | $\mathbf{1.22 \times 10^2_{\pm 0.1 \times 10^1}}$ | $75.1_{\pm 0.3}$ | $49.3_{\pm 1.0}$ | $26.2_{\pm 0.8}$ | $50.9_{\pm 0.3}$ |
| | Mult | $\mathbf{8.85 \times 10^4_{\pm 2.0 \times 10^3}}$ | $\mathbf{1.22 \times 10^2_{\pm 0.1 \times 10^1}}$ | $74.2_{\pm 0.3}$ | $\mathbf{52.9_{\pm 0.9}}$ | $29.9_{\pm 0.8}$ | $\mathbf{53.0_{\pm 0.3}}$ |

- Using an ETF classifier as the linear layer and training only the feature extractor with WD and CE (WD&ETF).

- Using an ETF classifier as the linear layer and training only the feature extractor with WD, FR, and CE (WD&FR&ETF).

As for the bottom three methods, we also compared with additive LA (Menon et al., 2020), with multiplicative LA (Kim & Kim, 2020), and without LA. To ensure that the model correctly classifies both samples of the head classes and tail classes, we consider the average of the per-class accuracy of all classes and each group, i.e., *Many*, *Medium*, and *Few*.

Table 5 lists the FDRs and accuracy of each method for CIFAR100-LT. We show the results on the other datasets including tabular data (CIFAR10-LT, mini-ImageNet-LT, ImageNet-LT, and Helena) and the other model (ResNeXt50) in Appendix A.5. First, training with the ETF classifier in addition to WD increases the FDR for both training and test features. Using FR further improves the FDR. However, it does not boost the average accuracy much. In contrast, LA enhances the classification accuracy of *Medium* and *Few* classes in all cases; thus, it also increase the final average accuracy significantly. It was observed that multiplicative LA improved accuracy to a greater extent than additive LA and that WD&FR&ETF&Multiplicative LA outperformed WB on average accuracy despite our method requiring only one-stage training.

## 6 CONCLUSION

We theoretically and experimentally demonstrated how WB improves accuracy in LTR by investigating each components of WB. In the first stage of training, WD and CE cause following three effects, decrease in inter-class cosine similarity, reduction in the scaling parameters of BN, and improvement of features in the ease of increasing FDR, which enhance the FDR of the features. In the second stage, WD, CB, and the norm of features that increased in the tail classes work as multiplicative LA, by improving the classification accuracy of the tail classes. Our analysis also reveals a training method that achieves higher accuracy than WB by improving each stage based on the objectives. This method is simple, thus we recommend trying it first. A limitation is that our experiments were limited to MLP, ResNet, and ResNeXt and the usefulness in other models such as ViT (Dosovitskiy et al., 2021) is unknown. Future research will include experiments and development of useful methods for such models.

## ACKNOWLEDGEMENTS

We would like to express my sincere gratitude to the members of Issei Sato Laboratory for their invaluable discussions. We are also grateful to the reviewers for their diligent critique, which has significantly improved the quality of this paper. This work was supported by JSPS KAKENHI Grant Number 20H04239 Japan.

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

# A  APPENDIX

## A.1  ACRONYM AND NOTATION

Tables 6 and 7 summarizes the abbreviation and notations used in this paper, respectively.

Table 6: Table of abbreviation.

| Abbreviation | Definition |
|---|---|
| BN | batch normalization (Ioffe & Szegedy, 2015) |
| CB | class-balanced loss (Cui et al., 2019) |
| CE | cross entropy |
| DNN | deep neural network |
| ETF | equiangular tight frame |
| FDR | Fisher's discriminant ratio (Fisher, 1936) |
| FR | feature regularization |
| LA | logit adjustment (Kim & Kim, 2020; Menon et al., 2020) |
| LTR | long-tailed recognition |
| MLP | multilayer perceptron |
| NC | neural collapse (Papyan et al., 2020) |
| ResBlock | residual block (He et al., 2016) |
| SELI | simplex-encoded-labels interpolation |
| SGD | stochastic gradient descent |
| WB | weight balancing (Alshammari et al., 2022) |
| WD | weight decay |

Table 7: Table of notations.

| Variable | Definition |
|---|---|
| $C$ | number of classes |
| $d$ | number of dimensions for features |
| $N$ / $N_k$ | number of samples / of class $k$ |
| $\overline{N}$ | harmonic mean of number of samples per class |
| $\lambda, \zeta$ | hyper parameter of weight decay / feature regularization |
| $\rho$ | imbalance factor: $\frac{\max_k N_k}{\min_k N_k}$ |
| $\mathcal{X}$/ $\mathcal{Y}$ | domain of samples / labels |
| $\mathbf{x}$ / $y$ / $\mathbf{z}$ | sample / label / logit |
| $\mathcal{D}$/$\mathcal{D}_k$ | dataset of all classes / class $k$ |
| $\boldsymbol{f}$ / $\boldsymbol{g}$ / $\boldsymbol{h}$ | neural network of whole model / feature extractor / linear classifier |
| $\boldsymbol{g}(\mathbf{x})$ / $\overline{\boldsymbol{g}}(\mathbf{x})$ | feature of sample $\mathbf{x}$ / normalized feature of sample $\mathbf{x}$ |
| $\boldsymbol{\mu}$ / $\boldsymbol{\mu_k}$ | mean of inner-class mean of features / inner-class mean of features for class $k$ |
| $\boldsymbol{\Theta}$ / $\boldsymbol{\Theta}_g$ / $\boldsymbol{\Theta}_h$ | set of parameters of $\boldsymbol{f}$ / $\boldsymbol{g}$ / $\boldsymbol{h}$ |
| $\boldsymbol{\theta}$ | parameter in $\boldsymbol{\Theta}$ |
| $\mathbf{W}$ / $\mathbf{w}_k$ | linear layers' weight matrix / vector of class $k$ |
| $\ell$ / $\ell_{\mathrm{CE}}$ / $\ell_{\mathrm{CB}}$ | loss function / of CE / of CB |
| $F$ / $F_{\mathrm{WB}}$ | objective function / in the second training stage of WB |
| $\cos(\cdot, \cdot)$ | cosine similarity of two vectors |
| $\tau$ / $\gamma$ | hyperparameter of additive LA / multiplicative LA |

## A.2  RELATED WORK

### A.2.1  LONG-TAILED RECOGNITION

There are three main approaches to LTR; "Class Re-balancing, Information Augmentation, and Module Improvement" (Zhang et al., 2021). "Class Re-balancing" adjusts the imbalance in the number of samples per class at various stages to prevent accuracy deterioration. It includes logit

Table 8: Comparison of existing LTR methods with our simplified method. A symbol ✓ indicates that the component is required, - indicates that the component is not required. We regard the deferred re-balancing optimization schedule (Cao et al., 2019) as two-stage training. Our method does not require complex technique like many other methods.

| Components \ Methods | Kim & Kim (2020) | Menon et al. (2020) | Yang et al. (2022) | Ma et al. (2022) | Liu et al. (2023) | Alshammari et al. (2022) | Cao et al. (2019) | Kang et al. (2023) | Li et al. (2022) | Long et al. (2022) | Ma et al. (2021) | Tian et al. (2022) | Ours |
|---|---|---|---|---|---|---|---|---|---|---|---|---|---|
| Number of training stages | 1 | 1 | 1 | 3 | 2 | 2 | 2 | 2 | 2 | 1* | 2 | 2* | 1 |
| Devising loss functions | - | - | ✓ | ✓ | - | ✓ | ✓ | ✓ | ✓ | ✓ | ✓ | ✓ | - |
| Resampling | - | - | - | - | ✓ | - | - | ✓ | ✓ | - | ✓ | ✓ | - |
| Contrastive learning | - | - | - | - | - | - | - | ✓ | ✓ | - | ✓ | ✓ | - |
| Training of linear layer | ✓ | ✓ | - | ✓ | ✓ | ✓ | ✓ | ✓ | ✓ | ✓ | -[2] | ✓[3] | - |
| Extra text encoder | - | - | - | - | - | - | - | - | - | ✓ | ✓ | ✓ | - |
| Extra image encoder | - | - | - | - | - | - | - | - | - | - | ✓ | - | - |
| Extra dataset | - | - | - | - | - | - | - | - | - | - | - | ✓ | - |

adjustment (LA) (Kim & Kim, 2020; Menon et al., 2020) and balancing the loss function such as CB and dynamic semantic-scale-balanced loss (Ma et al., 2022). "Information Augmentation" prevents the accuracy from degrading by supplementing the information of the tail classes that lacks the number of samples (Chu et al., 2020; Liu et al., 2020; Wang et al., 2021). "Module Improvement" improves accuracy by increasing the performance of each module of the network individually, e.g., training feature extractors and classifiers separately (Kang et al., 2020), fixing the linear layer to the ETF classifier for the better feature extractor (Yang et al., 2022), and regularizing to occur NC (Liu et al., 2023). In recent years, as Kang et al. (2023) found contrastive learning (Hadsell et al., 2006) is effective for imbalanced data, many methods have been proposed to use it (Kang et al., 2023; Li et al., 2022; Ma et al., 2021; Tian et al., 2022). Ma et al. (2021), Long et al. (2022), and Tian et al. (2022) also include vision-language models such as CLIP (Radford et al., 2021) and leverage text features to improve accuracy. However, these usually have problems such as slow convergence and complex models (Liu et al., 2023). WB is a combination of "Class Re-balancing" and "Module Improvement". Table 8 compares our simplification of WB with typical existing methods, including those that have achieved SOTA. While many methods use complex innovations, we aimed to improve on a simple structure.

**Two-stage learning**  Two-stage learning (Kang et al., 2020) is a method for improving accuracy by dividing LTR training into two stages: feature extractor training and classifier training. It is used in numerous methods (Cao et al., 2019; Ma et al., 2021; 2022; Li et al., 2022; Tian et al., 2022; Liu et al., 2023; Kang et al., 2023), including WB, because of its simple but significant improvement in accuracy. Note that since our work analyzes WB, the formulation of two-stage learning is based on WB. For example, the classifier weights are initialized randomly at the start of the second stage of training in Kang et al. (2020) but not in WB.

---

[2]This method does not require linear layers.
[3]This method requires the LGR head to be trained (Tian et al., 2022).

## A.3 PROOF

### A.3.1 PROOF OF THEOREM 1

*Proof.* Define $p_k(\boldsymbol{g}(\mathbf{x}_i))$ as $\frac{\exp(\mathbf{w}_k^\top \boldsymbol{g}(\mathbf{x}_i))}{\sum_{l=1}^C \exp(\mathbf{w}_l^\top \boldsymbol{g}(\mathbf{x}_i))}$. For $\mathbf{x} \in \{\mathbf{x}_i, \mathbf{x}_j\}$, we have

$$
\begin{aligned}
\left\| \frac{\partial \ell_{\mathrm{CE}}}{\partial \boldsymbol{g}(\mathbf{x})} \right\|_2 &= \left\| \frac{\partial \ell_{\mathrm{CE}}}{\partial \boldsymbol{g}(\mathbf{x})} \right\|_2 \|\mathbf{w}_y\|_2 \geq \left| \left\langle \frac{\partial \ell_{\mathrm{CE}}}{\partial \boldsymbol{g}(\mathbf{x})}, \mathbf{w}_y \right\rangle \right| \\
&= \left| -(1 - p_y(\boldsymbol{g}(\mathbf{x})))\mathbf{w}_y^\top \mathbf{w}_y + \sum_{l \neq y} p_l(\boldsymbol{g}(\mathbf{x}))\mathbf{w}_l^\top \mathbf{w}_y \right| \\
&= \left| -(1 - p_y(\boldsymbol{g}(\mathbf{x}))) - \frac{1}{C-1} \sum_{l \neq y} p_l(\boldsymbol{g}(\mathbf{x})) \right| = \frac{C}{C-1}(1 - p_y(\boldsymbol{g}(\mathbf{x}))) \\
&\geq (1 - p_y(\boldsymbol{g}(\mathbf{x}))).
\end{aligned}
\tag{4}
$$

Therefore,

$$
\begin{aligned}
(1 - p_y(\boldsymbol{g}(\mathbf{x}))) &\leq \left\| \frac{\partial \ell_{\mathrm{CE}}}{\partial \boldsymbol{g}(\mathbf{x})} \right\|_2 \leq \epsilon \\
&\Rightarrow p_y(\boldsymbol{g}(\mathbf{x})) \geq 1 - \epsilon.
\end{aligned}
\tag{5}
$$

For $p_y(\boldsymbol{g}(\mathbf{x}))$, using Jensen's inequality, we get

$$
\begin{aligned}
p_y(\boldsymbol{g}(\mathbf{x})) &= \frac{\exp(\mathbf{w}_y^\top \boldsymbol{g}(\mathbf{x}))}{\sum_{l=1}^C \exp\left(\mathbf{w}_l^\top \boldsymbol{g}(\mathbf{x})\right)} = \frac{\exp(\mathbf{w}_y^\top \boldsymbol{g}(\mathbf{x}))}{\exp(\mathbf{w}_y^\top \boldsymbol{g}(\mathbf{x}) + (C-1) \sum_{l \neq y} \frac{1}{C-1} \exp\left(\mathbf{w}_l^\top \boldsymbol{g}(\mathbf{x})\right)} \\
&\leq \frac{\exp(\mathbf{w}_y^\top \boldsymbol{g}(\mathbf{x}))}{\exp(\mathbf{w}_y^\top \boldsymbol{g}(\mathbf{x}) + (C-1) \exp\left(\left(\frac{1}{C-1} \sum_{l \neq y} \mathbf{w}_l\right)^\top \boldsymbol{g}(\mathbf{x})\right)} \\
&= \frac{\exp(\mathbf{w}_y^\top \boldsymbol{g}(\mathbf{x}))}{\exp(\mathbf{w}_y^\top \boldsymbol{g}(\mathbf{x}) + (C-1) \exp\left(-\frac{1}{C-1}\mathbf{w}_y^\top \boldsymbol{g}(\mathbf{x})\right)} \\
&= \frac{1}{1 + (C-1) \exp\left(-\frac{C}{C-1}\mathbf{w}_y^\top \boldsymbol{g}(\mathbf{x})\right)}.
\end{aligned}
\tag{6}
$$

Define $\overline{\boldsymbol{g}}(\mathbf{x}_i)$ as $\frac{\boldsymbol{g}(\mathbf{x}_i)}{\|\boldsymbol{g}(\mathbf{x}_i)\|_2}$. By (5) and (6), we have

$$
\begin{aligned}
(C-1) \exp\left(-\frac{C}{C-1}\mathbf{w}_y^\top \boldsymbol{g}(\mathbf{x})\right) &\leq \frac{\epsilon}{1-\epsilon} \\
\Rightarrow \mathbf{w}_y^\top \overline{\boldsymbol{g}}(\mathbf{x}) &\geq \frac{1}{\|\boldsymbol{g}(\mathbf{x})\|_2} \frac{C-1}{C} \log\left(\frac{(C-1)(1-\epsilon)}{\epsilon}\right) \\
\Rightarrow \mathbf{w}_y^\top \overline{\boldsymbol{g}}(\mathbf{x}) &\geq \frac{1}{L} \frac{C-1}{C} \log\left(\frac{(C-1)(1-\epsilon)}{\epsilon}\right) \equiv \delta.
\end{aligned}
\tag{7}
$$

Note that $\epsilon \leq \frac{1}{C}$, $L \leq 2\sqrt{2} \log\left(C-1\right) < 2\sqrt{2} \log\left(C-1\right)\frac{C}{C-1}$, which means $\delta > \frac{1}{\sqrt{2}}$. Also, $\delta$ is a lower bound of the inner product of the two unit vectors. Therefore, we have

$$
\delta \in \left(\frac{1}{\sqrt{2}}, 1\right].
\tag{8}
$$

Let $\angle(\mathbf{s}, \mathbf{t}) \equiv \arccos(\cos(\mathbf{s}, \mathbf{t}))$, which represents the angle between vectors $\mathbf{s}$ and $\mathbf{t}$. For example, we have $\angle(\mathbf{w}_{y_i}, \mathbf{w}_{y_j}) = \arccos\left(-\frac{1}{C-1}\right)$. Using (8), the following holds.

$$
\begin{aligned}
\angle(\overline{\boldsymbol{g}}(\mathbf{x}_i), \overline{\boldsymbol{g}}(\mathbf{x}_j)) &\geq \angle(\mathbf{w}_{y_i}, \mathbf{w}_{y_j}) - \angle(\mathbf{w}_{y_i}, \overline{\boldsymbol{g}}(\mathbf{x}_i)) - \angle(\mathbf{w}_{y_j}, \overline{\boldsymbol{g}}(\mathbf{x}_j)) \\
&\geq \angle(\mathbf{w}_{y_i}, \mathbf{w}_{y_j}) - 2 \max_{\mathbf{x} \in \{\mathbf{x}_i, \mathbf{x}_j\}} (\angle(\mathbf{w}_y, \overline{\boldsymbol{g}}(\mathbf{x}))) \\
&\geq 0.
\end{aligned}
\tag{9}
$$

Rewrite $\angle(\mathbf{w}_{y_i}, \mathbf{w}_{y_j})$ and $\max_{\mathbf{x} \in \{\mathbf{x}_i, \mathbf{x}_j\}}(\angle(\mathbf{w}_y, \overline{\boldsymbol{g}}(\mathbf{x}_x)))$ as $\alpha$ and $\beta$ respectively. Since the cosine is monotonically decreasing in $[0, \pi]$, we get

$$
\begin{aligned}
\overline{\boldsymbol{g}}(\mathbf{x}_i)^\top \overline{\boldsymbol{g}}(\mathbf{x}_j) &\leq \cos(\alpha - 2\beta) \\
&= \cos\alpha(2\cos^2\beta - 1) + 2\sin\alpha\sin\beta\cos\beta \\
&\leq \cos\alpha(2\cos^2\beta - 1) + 2\sqrt{1 - \cos^2\beta}\cos\beta
\end{aligned}
\tag{10}
$$

Note that $x\sqrt{1 - x^2}$ is monotonically decreasing and $2x^2 \geq 1$ holds in $\left[\frac{1}{\sqrt{2}}, 1\right]$. Therefore, from (7), (8), and (10), the following holds.

$$
\cos(\boldsymbol{g}(\mathbf{x}_i), \boldsymbol{g}(\mathbf{x}_j)) \leq 2\delta\sqrt{1 - \delta^2}.
\tag{11}
$$

$\square$

### A.3.2 PROOF OF THEOREM 2

Define $\hat{\mathbf{w}}_k$ be $\frac{\overline{N}}{\lambda N}\boldsymbol{\mu}_k$. We prove a more strict theorem. You can easily derive Theorem 2 from the following theorem.

**Theorem 3.** *For any $k \in \mathcal{Y}$, if there exists $\mathbf{w}_k^*$ s.t. $\left.\frac{\partial F_{\mathrm{WB}}}{\partial \mathbf{w}_k}\right|_{\mathbf{w}_k = \mathbf{w}_k^*} = \mathbf{0}$ and $\|\mathbf{w}_k^*\|_2 = O\left(\frac{1}{\lambda\rho C}\right)$, we have*

$$
\forall k \in \mathcal{Y}, \|\mathbf{w}_k^* - \hat{\mathbf{w}}_k\|_2 \leq \frac{\overline{N}}{\lambda N}\|\boldsymbol{\mu}\|_2 + O\left(\frac{1}{\lambda^2\rho^2 C^2}\right).
\tag{12}
$$

Before proving the theorem, we first present the following lemmas.

**Lemma 1.** *For the dataset of Imbalance rate $\rho$, the following holds.*

$$
\frac{\overline{N}}{N} = O\left(\frac{1}{\rho C}\right).
\tag{13}
$$

*Proof.* Since $N_k = n\rho^{-\frac{k-1}{C-1}}$ holds, the harmonic mean and the number of all samples are

$$
\overline{N} = \frac{C}{\sum_{k=1}^{C} \frac{1}{N_k}} = \frac{nC}{\sum_{k=1}^{C} \rho^{\frac{k-1}{C-1}}},
\tag{14}
$$

$$
N = \sum_{k=1}^{C} N_k = n\sum_{k=1}^{C} \rho^{-\frac{k-1}{C-1}} = \frac{n\sum_{k=1}^{C} \rho^{\frac{k-1}{C-1}}}{\rho}.
\tag{15}
$$

Therefore,

$$
\frac{\overline{N}}{N} = \frac{\rho C}{\left(\sum_{k=1}^{C} \rho^{\frac{k-1}{C-1}}\right)^2} = \frac{\rho C\left(\rho^{\frac{1}{C-1}} - 1\right)^2}{\left(\rho^{\frac{C}{C-1}} - 1\right)^2}.
\tag{16}
$$

From (16), $\frac{\overline{N}}{N} = O\left(\frac{1}{\rho}\right)$ $(\rho \to \infty)$ is obvious. In addition, the following holds.

$$\lim_{C \to \infty} \frac{\overline{N}C}{N} = \lim_{C \to \infty} \frac{\rho C^2}{\left(\sum_{k=1}^{C} \rho^{\frac{k-1}{C-1}}\right)^2} \leq \lim_{C \to \infty} \frac{\rho C^2}{C^2} = \rho. \tag{17}$$

Thus, $\frac{\overline{N}}{N} = O\left(\frac{1}{C}\right)$ $(C \to \infty)$ is also satisfied.

$\square$

**Lemma 2.** *When* $\mathbf{w}_k = \frac{\overline{N}}{\lambda N}\boldsymbol{\mu}_k$ *is satisfied, we have*

$$\forall k \in \mathcal{Y}, \; \frac{\partial F_{\mathrm{WB}}}{\partial \mathbf{w}_k} \leq \frac{\overline{N}}{N}\boldsymbol{\mu} + O\left(\frac{1}{\lambda \rho^2 C^2}\right). \tag{18}$$

*Proof of Lemma 2.* For any $k \in \mathcal{Y}$, it holds that

$$\frac{\partial F_{\mathrm{WB}}}{\partial \mathbf{w}_k} = \frac{\overline{N}}{N} \sum_{j=1}^{C} \left( \frac{1}{N_j} \sum_{(\mathbf{x}_i,j) \in \mathcal{D}_j} \frac{\exp(\mathbf{w}_k^\top \boldsymbol{g}(\mathbf{x}_i))}{\sum_{l=1}^{C} \exp\left(\mathbf{w}_l^\top \boldsymbol{g}(\mathbf{x}_i)\right)} \boldsymbol{g}(\mathbf{x}_i) \right) - \frac{\overline{N}}{N}\boldsymbol{\mu}_k + \lambda \mathbf{w}_k. \tag{19}$$

In addition, using Lemma 1, we derive the following:

$$\frac{\exp(\mathbf{w}_k^\top \boldsymbol{g}(\mathbf{x}_i))}{\sum_{l=1}^{C} \exp\left(\mathbf{w}_l^\top \boldsymbol{g}(\mathbf{x}_i)\right)} = \frac{1}{C} + \left( \frac{\exp(\mathbf{w}_k^\top \boldsymbol{g}(\mathbf{x}_i))}{\sum_{l=1}^{C} \exp\left(\mathbf{w}_l^\top \boldsymbol{g}(\mathbf{x}_i)\right)} - \frac{1}{C} \right)$$

$$\leq \frac{1}{C} + \left( \frac{1 + O(\mathbf{w}_k^\top \boldsymbol{g}(\mathbf{x}_i))}{\sum_{l=1}^{C} \left(1 + O(\mathbf{w}_l^\top \boldsymbol{g}(\mathbf{x}_i))\right)} - \frac{1}{C} \right)$$

$$= \frac{1}{C} + \frac{1}{C} \left( \left(1 + O\left(\frac{1}{\lambda \rho C}\right)\right) \left(1 - O\left(\frac{1}{\lambda \rho C^2}\right)\right) - 1 \right)$$

$$= \frac{1}{C} + O\left(\frac{1}{\lambda \rho C^2}\right). \tag{20}$$

Using this, the assumption and (19), we get

$$\frac{\partial F_{\mathrm{WB}}}{\partial \mathbf{w}_k} \leq \frac{\overline{N}}{N}\boldsymbol{\mu} + O\left(\frac{1}{\lambda \rho^2 C^2}\right). \tag{21}$$

$\square$

*Proof of Theorem 3.*

$$\left.\frac{\partial F_{\mathrm{WB}}}{\partial \mathbf{w}_k}\right|_{\mathbf{w}_k=\hat{\mathbf{w}}_k} = \left.\frac{\partial F_{\mathrm{WB}}}{\partial \mathbf{w}_k}\right|_{\mathbf{w}_k=\hat{\mathbf{w}}_k} - \left.\frac{\partial F_{\mathrm{WB}}}{\partial \mathbf{w}_k}\right|_{\mathbf{w}_k=\mathbf{w}_k^*}$$

$$= \lambda(\hat{\mathbf{w}}_k - \mathbf{w}_k^*)$$

$$+ \frac{\overline{N}}{N} \sum_{j=1}^{C} \left( \frac{1}{N_j} \sum_{(\mathbf{x}_i,j) \in \mathcal{D}_j} \left( \frac{\exp(\hat{\mathbf{w}}_k^\top \boldsymbol{g}(\mathbf{x}_i))}{\sum_{l=1}^{C} \exp\left(\hat{\mathbf{w}}_l^\top \boldsymbol{g}(\mathbf{x}_i)\right)} - \frac{\exp(\mathbf{w}_k^{*\top} \boldsymbol{g}(\mathbf{x}_i))}{\sum_{l=1}^{C} \exp\left(\mathbf{w}_l^{*\top} \boldsymbol{g}(\mathbf{x}_i)\right)} \right) \boldsymbol{g}(\mathbf{x}_i) \right). \tag{22}$$

Here, similar to (20), the following can be derived;

$$\frac{\exp(\hat{\mathbf{w}}_k^\top \boldsymbol{g}(\mathbf{x}_i))}{\sum_{l=1}^{C} \exp\left(\hat{\mathbf{w}}_l^\top \boldsymbol{g}(\mathbf{x}_i)\right)} - \frac{\exp(\mathbf{w}_k^{*\top} \boldsymbol{g}(\mathbf{x}_i))}{\sum_{l=1}^{C} \exp\left(\mathbf{w}_l^{*\top} \boldsymbol{g}(\mathbf{x}_i)\right)} = \frac{1}{C} - \frac{1}{C} + O\left(\frac{1}{\lambda \rho C^2}\right) = O\left(\frac{1}{\lambda \rho C^2}\right). \tag{23}$$

Using (22) and (23),

$$
\begin{aligned}
\lambda(\hat{\mathbf{w}}_k - \mathbf{w}_k^*) &= \left. \frac{\partial F_{\mathrm{WB}}}{\partial \mathbf{w}_k} \right|_{\mathbf{w}_k = \hat{\mathbf{w}}_k} + O\left( \frac{1}{\lambda \rho^2 C^2} \right) \\
&\leq \frac{\overline{N}}{N} \boldsymbol{\mu} + O\left( \frac{1}{\lambda \rho^2 C^2} \right).
\end{aligned}
\tag{24}
$$

Therefore,

$$
\| \mathbf{w}_k^* - \hat{\mathbf{w}}_k \|_2 \leq \frac{\overline{N}}{\lambda N} \| \boldsymbol{\mu} \|_2 + O\left( \frac{1}{\lambda^2 \rho^2 C^2} \right).
\tag{25}
$$

$\square$

## A.4 Settings

Our implementation is based on Alshammari et al. (2022) and Vigneswaran et al. (2021). For comparison, many of the experimental settings also follow Alshammari et al. (2022).

### A.4.1 Datasests

We created validation datasets from the portions of the training datasets because CIFAR10 and CIFAR100 have only training and test data. As with Liu et al. (2019), only 20 samples per class were taken from the training dataset to compose the validation dataset, and the training dataset was composed of the rest of the data. We set $N_1$ to 4980 for CIFAR10 and 480 for CIFAR100. In our experiments, we set imbalance factor $\rho$ to 100. We call the class $k$ *Many* if the number of training samples satisfies $1000 < N_k \leq 4980$ (resp. $100 < N_k \leq 480$), *Medium* if the number of training samples fullfills $200 \leq N_k \leq 1000$ (resp. $20 \leq N_k \leq 100$), and *Few* otherwise for CIFAR10-LT (resp. CIFAR100-LT). For mini-ImageNet-LT and ImageNet-LT, the same applies as for CIFAR100-LT.

### A.4.2 Models

As for MLPs, one module block consists of three layers: a linear layer outputting 1024 dimensional features, a BN layer, and a ReLU layer. The layers are stacked in sequence and the blocks composed of them are as well. As for ResBlocks, each block has the same structure as He et al. (2016), except that the linear layer outputs 1024 dimensional features. These blocks are combined into a sequential, and at the bottom of it, we further concatenate an MLP block. In other words, the input first passes through the linear layer, BN, and ReLU before flowing into the residual blocks. In both cases, a classifier consisting of a linear layer and a softmax activation layer is on the top and does not count as one block. For example, in MLP3, the features pass through the three blocks and the classifier in sequence.

### A.4.3 Evaluation metrics

Unless otherwise noted, we used the following values for hyperparameters for the ResNet. The optimizer was SGD with momentum = 0.9 and cosine learning rate scheduler (Loshchilov & Hutter, 2017) to gradually decrease the learning rate from 0.01 to 0. The batch size was 64, and the number of epochs was 320 for the first stage and 10 for the second stage. As loss functions, we used naive CE and CB. As for CB, we used class-balanced CE with $\beta = 0.9999$. We set $\lambda$ of WD to 0.005 in the first stage and 0.1 in the second stage. For mini-ImageNet-LT, we set $\lambda$ for the first stage to 0.003. We set $\zeta$ of FR to 0.01. We calculated the MaxNorm's threshold $\eta$ as in Alshammari et al. (2022). We searched the optimal $\tau$ and $\gamma$ for the LA by cross-validation using the validation data. We chose the optimal value of $\tau$ from $\{1.00, 1.05, \ldots, 2.00\}$ and $\gamma$ from $\{0.00, 0.05, \ldots, , 1.00\}$.

For ImageNet-LT, We searched hyperparameters using the validation dataset for the ones not published in Alshammari et al. (2022) and used the following values. We reduced the learning rate gradually from 0.05 to 0. The number of epochs was 200 for the first stage. We set $\lambda$ of WD to 0.00024 in the first stage and 0.00003 in the second stage. We set $\zeta$ of FR to 0.0001. The other hyperparameters were the same as in CIFAR100-LT.

Table 9: FDRs and accuracy of models trained with an ETF classifier for CIFAR100-LT.

| Method | FDR | | Accuracy (%) | | | |
|---|---|---|---|---|---|---|
| | Train | Test | *Many* | *Medium* | *Few* | Average |
| WD&ETF | $3.33{\times}10^4_{\pm 2.3\times10^3}$ | $\mathbf{1.13{\times}10^2_{\pm 0.1\times10^1}}$ | $\mathbf{76.3_{\pm 0.3}}$ | $\mathbf{46.0_{\pm 0.4}}$ | $15.5_{\pm 0.6}$ | $\mathbf{46.9_{\pm 0.2}}$ |
| WD&ETF&DR | $\mathbf{8.25{\times}10^5_{\pm 7.0\times10^4}}$ | $9.46{\times}10^1_{\pm 0.8\times10^0}$ | $72.9_{\pm 0.2}$ | $43.0_{\pm 0.6}$ | $\mathbf{20.0_{\pm 0.9}}$ | $46.0_{\pm 0.4}$ |

We trained MLPs and ResBlocks with $\lambda$ set to $0.01$ and the number of epochs set to $150$. Other parameters were the same as above. FDRs and accuracy reported in our experiments were obtained by averaging the results from five training runs with different random seeds. We conducted experiments on an NVIDIA A100.

### A.4.4    WHY WE DO NOT USE DR LOSS

This section explains why we use ETF classifiers but not the dot-regression loss (DR) proposed in Yang et al. (2022). Table 9 compares FDRs and accuracy when we use WD and an ETF Classifier for the linear layer with two options, using CE or DR for the loss. This table shows that CE is superior in the test FDR and average accuracy. Theorem 1 also claims that training by CE and WD prevents the cone effect. It does not apply to DR, which is a squared loss function. Thus, it is difficult to prove whether DR is equally effective in preventing the cone effect.

### A.5    EXPERIMENTS

### A.5.1    EXPERIMENTS OF SEC.4.2

The right half of Figure 4 show the result of Sec.4.2 for mini-ImageNet-LT. We also examined the forget score (Toneva et al., 2019) of each method in Sec.4.2. Phenomena similar to the ones we showed in Sec.4.2 can be observed in the forgetting score of features trained with CB: Figure 3 shows that the forgetting score is higher on average with CB than with CE. In the absence of WD, this phenomenon is particularly evident for the *Many* classes, possibly because CB gives each image in *Many* classes less weight during training, which inhibits learning. These may be some of the reasons for the poor accuracy of CB compared with CE in training feature extractors reported by Kang et al. (2020).

### A.5.2    EXPERIMENTS OF SEC. 4.3

**High relative variance in shifting parameters of BN temporarily degrades FDR**    In Sec.4.3, we investigated the effect of the small scaling parameters of BN. What then is the impact of higher relative variance in the shifting parameters of BN? Our experiments have shown that they temporarily worsen the FDR in Figure 6. In the experiments, we trained MLPs with MNIST using three different methods: without WD (Naive), with WD (WD all), and with WD except for any shifting parameters of BN (WD w/o shift). We retrieved the output features of untrained and these trained models for each layer, applied ReLU to them, and examined their FDRs. We refer to these as ReLUFDRs. Figure 6 shows the increase and decrease in the ReLUFDRs of the intermediate outputs from the MLPs trained with each method. This figure indicates that the ReLUFDRs of the models trained without WD monotonically and gradually increases as the features pass through the layers. However, this is not the case for the models trained with WD. In particular, ReLUFDRs decrease drastically when we train the shifting parameters and apply the scaling and shifting parameters of the BN in the blocks close to the last layer. In this case, the ReLUFDRs increase when the features pass through the linear layer more than the ReLUFDRs decrease when the features pass through the BN later. These results suggest that applying WD to the scaling and shifting parameters of BN have a positive effect on training of the linear layer.

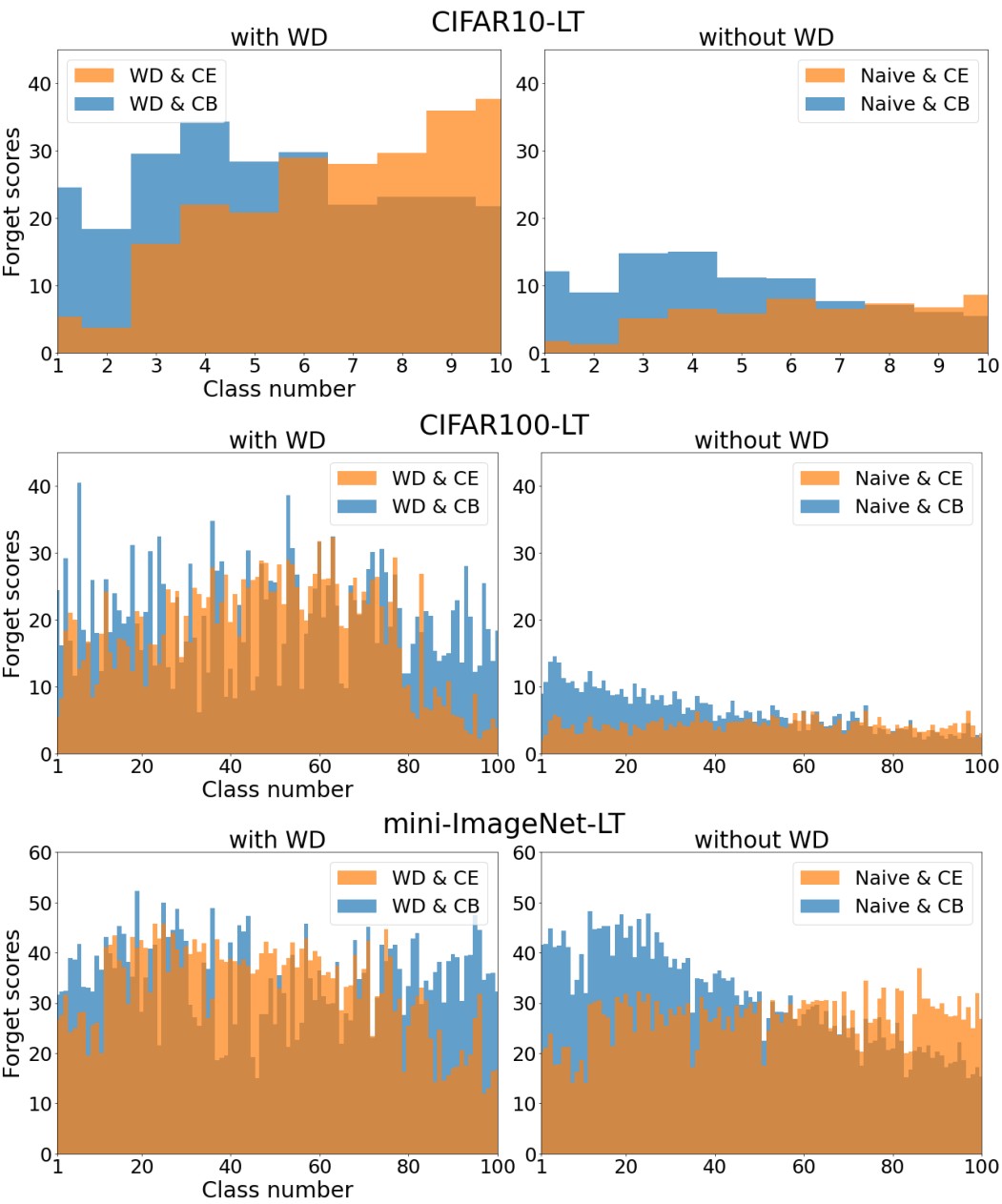

Figure 3: Average forgetting scores per class when models are trained with each method. These indicate higher forgetting scores when the models are trained with CB; this is particularly noticeable in the *Many* classes without WD.

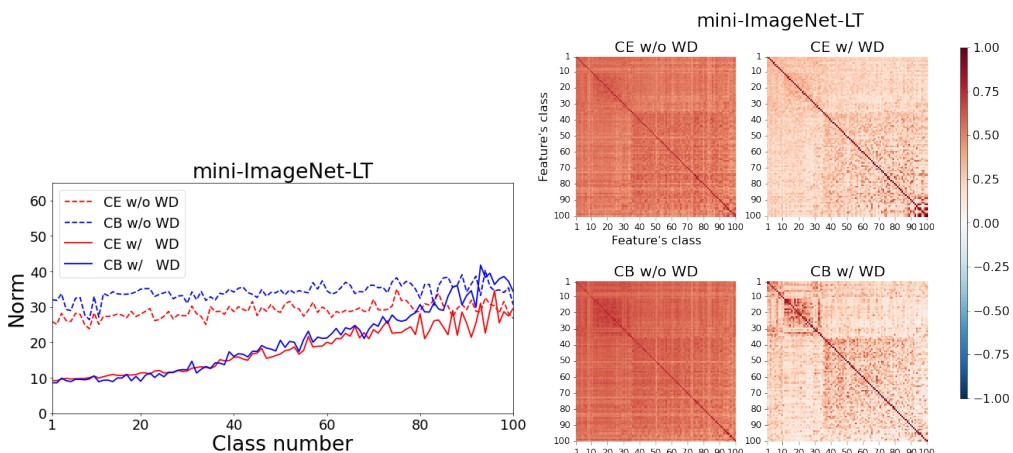

Figure 4: Results for mini-ImageNet-LT. (Left) Norm of mean per-class training features produced from models trained with each method. (Right) Heatmaps showing average cosine similarities of training features between two classes.

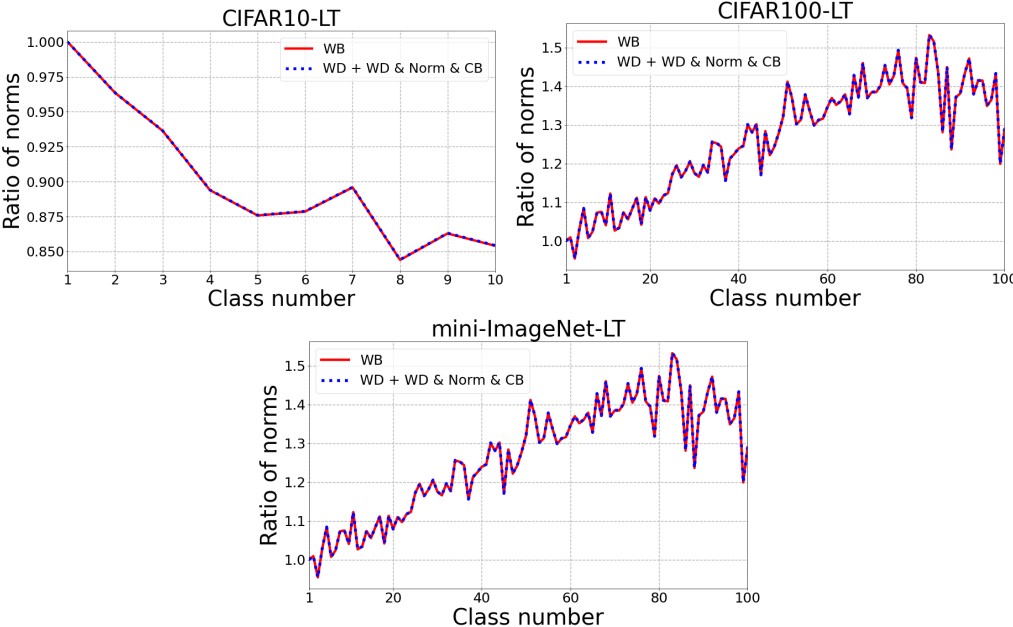

Figure 5: Norm ratio of mean per-class training features produced from models trained with each dataset and each method. Note that the vertical axis shows the ratio of the norm of the weights for each class with the one for the class of which sample size is the largest. Models trained with both methods have almost identical linear layer norms.

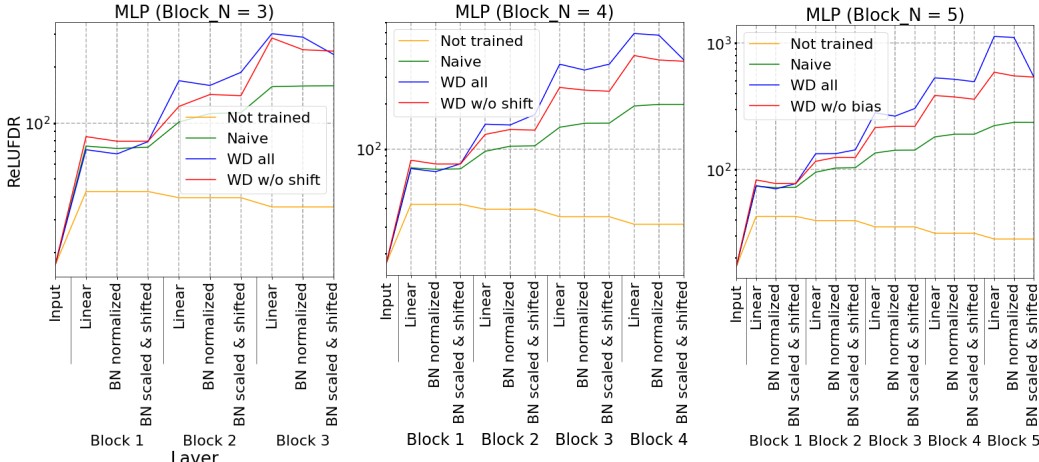

Figure 6: ReLUFDR of intermediate outputs of each MLP trained with each method. Training with WD decreases ReLUFDR significantly when features pass through the scaling and shifting layer in the BN of the final layer.

Table 10: Change in FDR as learned features pass through randomized linear layers and ReLUs multiple times. Red text indicates an increase compared to the FDR of the feature when the number of layers passed is one less, while blue text indicates a decrease.

| Method | Number of times through randomized linear layers | | | |
|---|---|---|---|---|
| | 0 | 1 | 2 | 3 |
| CE | $2.44 \times 10^2$ | $2.37 \times 10^2$ | $2.17 \times 10^2$ | $1.96 \times 10^2$ |
| WD w/o BN | $4.91 \times 10^2$ | $6.76 \times 10^2$ | $6.56 \times 10^2$ | $6.22 \times 10^2$ |
| WD | $6.10 \times 10^2$ | $2.47 \times 10^3$ | $2.34 \times 10^3$ | $1.60 \times 10^3$ |

### A.5.3 EXPERIMENTS OF SEC. 4.4

This subsection describes the experimental setup of Sec. 4.4 and additional experiments. We trained models for the balanced datasets using the same parameters settings as in Sec. 4.2. The FDRs of "before" are measured from the features obtained by models trained in the same way as the experiments in Sec. 4.2. These features are passed through an extra linear layer, and FDRs of them are measured as the "after". We initialized the weight $\left( \in \mathbb{R}^{d \times d} \right)$ and bias $\left( \in \mathbb{R}^{d \times 1} \right)$ of the extra linear layer with random values following a uniform distribution in $\left[ -\frac{1}{\sqrt{d}}, \frac{1}{\sqrt{d}} \right]$ and fixed them.

Table 10 shows the relationship between the FDRs and numbers of times features from MLP5 are applied by randomly initialized linear layers and ReLUs. The FDRs increase the first time, but gradually decrease the second and subsequent times.

### A.5.4 EXPERIMENTS OF SEC. 5.1

First, we experimentally show MaxNorm is not necessary for the second stage of WB. In the second training stage in WB, we observed that normalizing the norm of the weights to one before training (WD + WD & Norm & CB) instead of applying MaxNorm every epoch gives almost identical results as shown in Figure 5. This phenomenon is consistent with the fact that the regularization loses its effect beyond a certain epoch (Golatkar et al., 2019).

Figure 5 also shows that WB has difficulty in doing implicit LA when the number of classes is small. While the weights of the linear layer are larger for *Few* classes in the datasets with a sufficiently large number of classes, they remain lower for *Few* classes in CIFAR10-LT. These results are consistent with the conclusions drawn from Theorem 2.

Table 11: FDRs and accuracy of models trained with restricted WD and ETF Classifier for CIFAR100-LT.

| Method | LA | FDR | | Accuracy (%) | | | |
| | | Train | Test | *Many* | *Medium* | *Few* | Average |
|---|---|---|---|---|---|---|---|
| WD | N/A | $1.94\times10^2_{\pm0.8\times10^1}$ | $5.89\times10^1_{\pm1.0\times10^0}$ | **74.6**$_{\pm0.5}$ | 44.4$_{\pm0.7}$ | 13.4$_{\pm0.6}$ | 45.1$_{\pm0.5}$ |
| w/o BN | Add | $1.94\times10^2_{\pm0.8\times10^1}$ | $5.89\times10^1_{\pm1.0\times10^0}$ | 69.3$_{\pm0.8}$ | 49.4$_{\pm0.5}$ | **30.4**$_{\pm0.9}$ | **50.3**$_{\pm0.6}$ |
| & ETF | Mult | $1.94\times10^2_{\pm0.8\times10^1}$ | $5.89\times10^1_{\pm1.0\times10^0}$ | 67.6$_{\pm0.9}$ | **51.0**$_{\pm0.5}$ | 31.5$_{\pm1.3}$ | **50.6**$_{\pm0.7}$ |
| WD | N/A | $\mathbf{5.47\times10^4_{\pm8.0\times10^3}}$ | $\mathbf{1.05\times10^2_{\pm0.2\times10^1}}$ | **75.2**$_{\pm0.6}$ | 45.4$_{\pm0.9}$ | 16.2$_{\pm0.4}$ | 46.5$_{\pm0.1}$ |
| fixed BN | Add | $\mathbf{5.42\times10^4_{\pm8.0\times10^3}}$ | $\mathbf{1.05\times10^2_{\pm0.2\times10^1}}$ | 71.9$_{\pm0.6}$ | 44.8$_{\pm0.6}$ | 26.5$_{\pm0.5}$ | 48.4$_{\pm0.3}$ |
| &ETF | Mult | $\mathbf{5.42\times10^4_{\pm8.0\times10^3}}$ | $\mathbf{1.05\times10^2_{\pm0.2\times10^1}}$ | 72.2$_{\pm0.8}$ | 45.7$_{\pm0.5}$ | 26.9$_{\pm0.4}$ | 48.9$_{\pm0.3}$ |

Table 12: FDRs and accuracy of models trained with each method for CIFAR10-LT.

| Method | LA | FDR | | Accuracy (%) | | | |
| | | Train | Test | *Many* | *Medium* | *Few* | Average |
|---|---|---|---|---|---|---|---|
| CE | N/A | $8.17\times10^1_{\pm1.17\times10^1}$ | $2.17\times10^1_{\pm0.6\times10^0}$ | 87.7$_{\pm0.3}$ | 67.2$_{\pm3.1}$ | 47.4$_{\pm2.2}$ | 69.4$_{\pm1.0}$ |
| CB | N/A | $4.43\times10^1_{\pm1.21\times10^1}$ | $1.50\times10^1_{\pm0.9\times10^0}$ | 81.7$_{\pm1.3}$ | 60.6$_{\pm2.0}$ | 42.0$_{\pm4.7}$ | 63.5$_{\pm1.3}$ |
| | N/A | $2.97\times10^3_{\pm3.38\times10^3}$ | $4.21\times10^1_{\pm1.7\times10^0}$ | **89.1**$_{\pm1.8}$ | 76.6$_{\pm1.7}$ | 61.5$_{\pm5.0}$ | 77.1$_{\pm1.0}$ |
| WD | Add | $3.40\times10^3_{\pm4.32\times10^3}$ | $4.21\times10^1_{\pm1.7\times10^0}$ | 81.6$_{\pm3.8}$ | **79.5**$_{\pm1.2}$ | 79.7$_{\pm3.1}$ | **80.4**$_{\pm0.7}$ |
| | Mult | $3.40\times10^3_{\pm4.32\times10^3}$ | $4.21\times10^1_{\pm1.7\times10^0}$ | 86.2$_{\pm2.9}$ | **80.3**$_{\pm1.0}$ | 74.0$_{\pm3.9}$ | **80.8**$_{\pm0.5}$ |
| WB | N/A | $1.82\times10^3_{\pm2.58\times10^3}$ | $3.86\times10^1_{\pm4.5\times10^0}$ | **87.9**$_{\pm2.4}$ | 77.6$_{\pm1.5}$ | 67.9$_{\pm3.2}$ | 78.8$_{\pm0.6}$ |
| | N/A | $3.90\times10^4_{\pm4.57\times10^4}$ | $\mathbf{4.38\times10^1_{\pm3.2\times10^0}}$ | **89.7**$_{\pm1.9}$ | 73.9$_{\pm5.6}$ | 59.1$_{\pm1.8}$ | 75.8$_{\pm1.5}$ |
| WD&ETF | Add | $3.59\times10^4_{\pm4.18\times10^4}$ | $\mathbf{4.38\times10^1_{\pm3.2\times10^0}}$ | 87.2$_{\pm3.4}$ | 77.1$_{\pm5.1}$ | 72.5$_{\pm2.6}$ | **79.8**$_{\pm1.1}$ |
| | Mult | $3.59\times10^4_{\pm4.18\times10^4}$ | $\mathbf{4.38\times10^1_{\pm3.2\times10^0}}$ | 86.9$_{\pm4.2}$ | 79.0$_{\pm4.4}$ | 73.2$_{\pm2.9}$ | **80.4**$_{\pm0.8}$ |
| | N/A | $\mathbf{5.56\times10^5_{\pm4.29\times10^5}}$ | $\mathbf{4.78\times10^1_{\pm2.0\times10^0}}$ | **90.9**$_{\pm0.6}$ | 76.7$_{\pm1.4}$ | 56.2$_{\pm2.0}$ | 76.2$_{\pm0.4}$ |
| WD&FR &ETF | Add | $\mathbf{7.58\times10^5_{\pm6.06\times10^5}}$ | $\mathbf{4.78\times10^1_{\pm2.0\times10^0}}$ | 88.8$_{\pm1.1}$ | **78.7**$_{\pm1.2}$ | 70.3$_{\pm2.1}$ | **80.2**$_{\pm0.5}$ |
| | Mult | $\mathbf{7.58\times10^5_{\pm6.06\times10^5}}$ | $\mathbf{4.78\times10^1_{\pm2.0\times10^0}}$ | 89.9$_{\pm0.9}$ | **80.8**$_{\pm1.3}$ | 66.3$_{\pm2.1}$ | **80.1**$_{\pm0.4}$ |
| WD | N/A | $9.17\times10^1_{\pm3.60\times10^1}$ | $2.95\times10^1_{\pm2.8\times10^0}$ | 88.3$_{\pm1.4}$ | 75.6$_{\pm3.6}$ | 59.1$_{\pm1.8}$ | 75.7$_{\pm1.6}$ |
| w/o BN | Add | $9.26\times10^1_{\pm3.66\times10^1}$ | $2.95\times10^1_{\pm2.8\times10^0}$ | 85.9$_{\pm2.5}$ | **79.0**$_{\pm3.2}$ | 74.3$_{\pm1.5}$ | **80.3**$_{\pm1.5}$ |
| & ETF | Mult | $9.26\times10^1_{\pm3.66\times10^1}$ | $2.95\times10^1_{\pm2.8\times10^0}$ | 85.5$_{\pm2.5}$ | **79.0**$_{\pm3.2}$ | 75.8$_{\pm1.0}$ | **80.7**$_{\pm1.8}$ |
| WD | N/A | $7.02\times10^3_{\pm4.59\times10^3}$ | $4.29\times10^1_{\pm2.8\times10^0}$ | **89.5**$_{\pm1.9}$ | 74.3$_{\pm3.2}$ | 58.9$_{\pm5.8}$ | 75.7$_{\pm1.3}$ |
| fixed BN | Add | $6.60\times10^3_{\pm4.28\times10^3}$ | $4.29\times10^1_{\pm2.8\times10^0}$ | 79.3$_{\pm9.2}$ | **76.9**$_{\pm3.6}$ | 78.7$_{\pm5.4}$ | 78.4$_{\pm3.2}$ |
| &ETF | Mult | $6.60\times10^3_{\pm4.28\times10^3}$ | $4.29\times10^1_{\pm2.8\times10^0}$ | **84.5**$_{\pm6.9}$ | 78.5$_{\pm4.1}$ | 77.1$_{\pm5.6}$ | **80.5**$_{\pm2.6}$ |

### A.5.5 EXPERIMENTS OF SEC.5.2

We show the FDRs and accuracy of methods with restricted WD for CIFAR100-LT in Table 11. Table 12, 13, and Table 14 show the results for CIFAR10-LT, mini-ImageNet-LT, and ImageNet-LT respectively. We set $\zeta$ for CIFAR10-LT, mini-ImageNet-LT, and ImageNet-LT to 0.02, 0.001, and 0.0001 respectively.

We also experimented with the ResNeXt50 (Xie et al., 2017) as with Alshammari et al. (2022). We used the same hyperparameters as the ResNet. Table 15 presents the results for CIFAR100-LT.

Table 13: FDRs and accuracy of models trained with each method for mini-ImageNet-LT.

| Method | LA | FDR | | Accuracy (%) | | | |
|---|---|---|---|---|---|---|---|
| | | Train | Test | *Many* | *Medium* | *Few* | Average |
| CE | N/A | $7.56\times10^1_{\pm1.6\times10^0}$ | $4.28\times10^1_{\pm0.3\times10^0}$ | $72.1_{\pm0.5}$ | $34.1_{\pm0.6}$ | $18.3_{\pm0.5}$ | $42.7_{\pm0.2}$ |
| CB | N/A | $4.68\times10^1_{\pm0.7\times10^0}$ | $2.93\times10^1_{\pm0.2\times10^0}$ | $59.4_{\pm0.5}$ | $25.0_{\pm0.4}$ | $15.7_{\pm0.4}$ | $34.3_{\pm0.2}$ |
| WD | N/A | $6.58\times10^2_{\pm2.8\times10^1}$ | $1.01\times10^2_{\pm0.1\times10^1}$ | $\mathbf{81.7_{\pm0.4}}$ | $\mathbf{43.3_{\pm0.5}}$ | $20.3_{\pm0.7}$ | $49.8_{\pm0.3}$ |
| | Add | $6.51\times10^2_{\pm3.3\times10^1}$ | $1.01\times10^2_{\pm0.1\times10^1}$ | $80.5_{\pm0.3}$ | $42.1_{\pm0.6}$ | $39.3_{\pm0.2}$ | $\mathbf{54.7_{\pm0.3}}$ |
| | Mult | $6.51\times10^2_{\pm3.3\times10^1}$ | $1.01\times10^2_{\pm0.1\times10^1}$ | $79.9_{\pm0.2}$ | $41.6_{\pm0.8}$ | $40.3_{\pm0.3}$ | $\mathbf{54.6_{\pm0.4}}$ |
| WB | N/A | $6.52\times10^2_{\pm3.0\times10^1}$ | $1.00\times10^2_{\pm0.1\times10^1}$ | $80.1_{\pm0.2}$ | $\mathbf{43.9_{\pm0.7}}$ | $38.1_{\pm0.5}$ | $\mathbf{54.8_{\pm0.3}}$ |
| WD&ETF | N/A | $8.20\times10^2_{\pm6.5\times10^1}$ | $1.09\times10^2_{\pm0.2\times10^1}$ | $\mathbf{81.7_{\pm0.7}}$ | $42.9_{\pm0.4}$ | $21.6_{\pm1.2}$ | $50.1_{\pm0.2}$ |
| | Add | $8.27\times10^2_{\pm8.5\times10^1}$ | $1.09\times10^2_{\pm0.2\times10^1}$ | $79.6_{\pm0.8}$ | $43.0_{\pm0.6}$ | $37.5_{\pm0.7}$ | $54.1_{\pm0.2}$ |
| | Mult | $8.27\times10^2_{\pm8.5\times10^1}$ | $1.09\times10^2_{\pm0.2\times10^1}$ | $78.8_{\pm1.1}$ | $42.7_{\pm1.0}$ | $39.4_{\pm1.0}$ | $54.3_{\pm0.3}$ |
| WD&FR &ETF | N/A | $\mathbf{1.32\times10^3_{\pm1.2\times10^2}}$ | $1.20\times10^2_{\pm0.2\times10^1}$ | $81.7_{\pm0.6}$ | $43.1_{\pm0.7}$ | $20.8_{\pm0.7}$ | $49.9_{\pm0.5}$ |
| | Add | $\mathbf{1.31\times10^3_{\pm1.1\times10^2}}$ | $1.20\times10^2_{\pm0.2\times10^1}$ | $79.6_{\pm0.5}$ | $43.5_{\pm0.6}$ | $34.9_{\pm0.9}$ | $53.6_{\pm0.6}$ |
| | Mult | $\mathbf{1.31\times10^3_{\pm1.1\times10^2}}$ | $1.20\times10^2_{\pm0.2\times10^1}$ | $78.8_{\pm0.3}$ | $43.5_{\pm0.8}$ | $37.9_{\pm0.8}$ | $54.2_{\pm0.5}$ |
| WD w/o BN & ETF | N/A | $1.21\times10^2_{\pm0.1\times10^1}$ | $5.85\times10^1_{\pm0.3\times10^0}$ | $78.6_{\pm0.4}$ | $41.7_{\pm0.6}$ | $21.7_{\pm0.7}$ | $48.6_{\pm0.3}$ |
| | Add | $1.21\times10^2_{\pm0.1\times10^1}$ | $5.85\times10^1_{\pm0.3\times10^0}$ | $75.1_{\pm0.2}$ | $40.3_{\pm0.7}$ | $\mathbf{41.7_{\pm0.6}}$ | $52.9_{\pm0.3}$ |
| | Mult | $1.21\times10^2_{\pm0.1\times10^1}$ | $5.85\times10^1_{\pm0.3\times10^0}$ | $76.2_{\pm0.3}$ | $41.0_{\pm0.5}$ | $39.6_{\pm0.6}$ | $52.9_{\pm0.3}$ |
| WD fixed BN &ETF | N/A | $8.55\times10^2_{\pm2.0\times10^1}$ | $\mathbf{1.25\times10^2_{\pm0.1\times10^1}}$ | $80.6_{\pm0.2}$ | $42.8_{\pm0.5}$ | $20.3_{\pm0.5}$ | $49.3_{\pm0.1}$ |
| | Add | $8.56\times10^2_{\pm2.1\times10^1}$ | $\mathbf{1.25\times10^2_{\pm0.1\times10^1}}$ | $78.7_{\pm0.3}$ | $41.6_{\pm0.4}$ | $36.0_{\pm0.7}$ | $52.9_{\pm0.3}$ |
| | Mult | $8.56\times10^2_{\pm2.1\times10^1}$ | $\mathbf{1.25\times10^2_{\pm0.1\times10^1}}$ | $80.5_{\pm0.2}$ | $\mathbf{43.6_{\pm0.3}}$ | $25.6_{\pm0.5}$ | $51.1_{\pm0.1}$ |

Table 14: FDRs and accuracy of models trained with each method for ImageNet-LT.

| Method | LA | FDR | | Accuracy (%) | | | |
|---|---|---|---|---|---|---|---|
| | | Train | Test | *Many* | *Medium* | *Few* | Average |
| CE | N/A | $1.35\times10^2_{\pm0.2\times10^1}$ | $1.58\times10^2_{\pm0.2\times10^1}$ | $54.7_{\pm0.2}$ | $30.1_{\pm0.4}$ | $12.1_{\pm0.2}$ | $37.1_{\pm0.3}$ |
| CB | N/A | $1.00\times10^2_{\pm0.8\times10^1}$ | $1.34\times10^2_{\pm0.9\times10^1}$ | $47.9_{\pm2.4}$ | $24.1_{\pm2.0}$ | $8.19_{\pm1.1}$ | $31.1_{\pm2.0}$ |
| WD | N/A | $3.79\times10^2_{\pm0.8\times10^1}$ | $2.99\times10^2_{\pm0.3\times10^1}$ | $67.4_{\pm0.5}$ | $42.0_{\pm0.4}$ | $15.2_{\pm0.3}$ | $48.1_{\pm0.4}$ |
| | Add | $3.77\times10^2_{\pm0.8\times10^1}$ | $2.99\times10^2_{\pm0.3\times10^1}$ | $62.9_{\pm0.4}$ | $48.8_{\pm0.5}$ | $35.2_{\pm0.3}$ | $52.4_{\pm0.4}$ |
| | Mult | $3.77\times10^2_{\pm0.8\times10^1}$ | $2.99\times10^2_{\pm0.3\times10^1}$ | $62.8_{\pm0.4}$ | $\mathbf{49.3_{\pm0.6}}$ | $34.4_{\pm0.4}$ | $\mathbf{52.5_{\pm0.4}}$ |
| WB | N/A | $3.78\times10^2_{\pm0.8\times10^1}$ | $2.99\times10^2_{\pm0.3\times10^1}$ | $62.4_{\pm0.5}$ | $\mathbf{50.0_{\pm0.5}}$ | $31.1_{\pm0.5}$ | $52.2_{\pm0.4}$ |
| WD&ETF | N/A | $6.98\times10^2_{\pm1.1\times10^1}$ | $4.56\times10^2_{\pm0.7\times10^1}$ | $\mathbf{68.4_{\pm0.4}}$ | $43.6_{\pm0.4}$ | $17.7_{\pm0.7}$ | $49.6_{\pm0.4}$ |
| | Add | $6.95\times10^2_{\pm1.0\times10^1}$ | $4.56\times10^2_{\pm0.7\times10^1}$ | $64.0_{\pm0.3}$ | $\mathbf{49.6_{\pm0.3}}$ | $35.5_{\pm0.4}$ | $\mathbf{53.2_{\pm0.3}}$ |
| | Mult | $6.95\times10^2_{\pm1.0\times10^1}$ | $4.56\times10^2_{\pm0.7\times10^1}$ | $63.4_{\pm0.4}$ | $\mathbf{50.0_{\pm0.4}}$ | $35.1_{\pm0.5}$ | $\mathbf{53.2_{\pm0.3}}$ |
| WD&FR &ETF | N/A | $\mathbf{1.03\times10^3_{\pm0.4\times10^2}}$ | $\mathbf{5.46\times10^2_{\pm0.6\times10^1}}$ | $68.1_{\pm0.2}$ | $43.0_{\pm0.2}$ | $17.0_{\pm0.5}$ | $49.1_{\pm0.2}$ |
| | Add | $\mathbf{1.03\times10^3_{\pm0.4\times10^2}}$ | $\mathbf{5.46\times10^2_{\pm0.6\times10^1}}$ | $63.3_{\pm0.3}$ | $48.7_{\pm0.1}$ | $\mathbf{36.4_{\pm0.7}}$ | $52.6_{\pm0.2}$ |
| | Mult | $\mathbf{1.03\times10^3_{\pm0.4\times10^2}}$ | $\mathbf{5.46\times10^2_{\pm0.6\times10^1}}$ | $62.4_{\pm0.4}$ | $49.3_{\pm0.2}$ | $\mathbf{36.0_{\pm0.5}}$ | $52.6_{\pm0.2}$ |

Table 15: FDRs and accuracy of ResNeXt50 trained with each method for CIFAR100-LT.

| Method | LA | FDR | | Accuracy (%) | | | |
| | | Train | Test | *Many* | *Medium* | *Few* | Average |
|---|---|---|---|---|---|---|---|
| CE | N/A | $2.04{\times}10^2_{\pm0.6\times10^1}$ | $7.32{\times}10^1_{\pm1.1\times10^0}$ | $59.9_{\pm0.4}$ | $32.8_{\pm0.7}$ | $9.51_{\pm0.3}$ | $34.8_{\pm0.4}$ |
| CB | N/A | $1.37{\times}10^2_{\pm1.6\times10^1}$ | $5.19{\times}10^1_{\pm1.1\times10^0}$ | $44.8_{\pm1.4}$ | $20.7_{\pm0.9}$ | $4.62_{\pm0.4}$ | $23.9_{\pm0.9}$ |
| WD | N/A | $8.36{\times}10^4_{\pm7.4\times10^3}$ | $\mathbf{2.06{\times}10^2}_{\pm0.3\times10^1}$ | $\mathbf{77.9}_{\pm0.2}$ | $48.3_{\pm0.5}$ | $14.9_{\pm0.8}$ | $48.0_{\pm0.3}$ |
| | Add | $8.41{\times}10^4_{\pm7.9\times10^3}$ | $\mathbf{2.06{\times}10^2}_{\pm0.3\times10^1}$ | $73.4_{\pm0.5}$ | $48.6_{\pm0.8}$ | $\mathbf{32.9}_{\pm0.8}$ | $52.2_{\pm0.2}$ |
| | Mult | $8.41{\times}10^4_{\pm7.9\times10^3}$ | $\mathbf{2.06{\times}10^2}_{\pm0.3\times10^1}$ | $73.7_{\pm0.6}$ | $49.2_{\pm1.1}$ | $\mathbf{33.8}_{\pm1.5}$ | $52.8_{\pm0.2}$ |
| WB | N/A | $3.19{\times}10^5_{\pm2.0\times10^4}$ | $\mathbf{2.06{\times}10^2}_{\pm0.2\times10^1}$ | $\mathbf{77.5}_{\pm0.2}$ | $51.2_{\pm0.9}$ | $21.1_{\pm0.6}$ | $50.8_{\pm0.3}$ |
| WD&ETF | N/A | $1.67{\times}10^5_{\pm2.4\times10^4}$ | $\mathbf{2.02{\times}10^2}_{\pm0.3\times10^1}$ | $\mathbf{77.7}_{\pm0.6}$ | $48.8_{\pm0.9}$ | $17.4_{\pm0.5}$ | $48.9_{\pm0.4}$ |
| | Add | $1.67{\times}10^5_{\pm2.4\times10^4}$ | $\mathbf{2.02{\times}10^2}_{\pm0.3\times10^1}$ | $72.7_{\pm0.9}$ | $50.3_{\pm1.0}$ | $31.1_{\pm0.5}$ | $52.0_{\pm0.6}$ |
| | Mult | $1.67{\times}10^5_{\pm2.4\times10^4}$ | $\mathbf{2.02{\times}10^2}_{\pm0.3\times10^1}$ | $74.6_{\pm0.7}$ | $\mathbf{54.0}_{\pm1.0}$ | $30.9_{\pm0.9}$ | $\mathbf{53.8}_{\pm0.4}$ |
| WD&FR &ETF | N/A | $\mathbf{3.19{\times}10^5}_{\pm1.8\times10^4}$ | $\mathbf{2.06{\times}10^2}_{\pm0.2\times10^1}$ | $\mathbf{77.4}_{\pm0.3}$ | $49.8_{\pm0.5}$ | $18.2_{\pm0.4}$ | $49.4_{\pm0.3}$ |
| | Add | $\mathbf{3.17{\times}10^5}_{\pm1.8\times10^4}$ | $\mathbf{2.06{\times}10^2}_{\pm0.2\times10^1}$ | $76.3_{\pm0.3}$ | $51.9_{\pm0.4}$ | $26.2_{\pm0.4}$ | $52.2_{\pm0.2}$ |
| | Mult | $\mathbf{3.17{\times}10^5}_{\pm1.8\times10^4}$ | $\mathbf{2.06{\times}10^2}_{\pm0.2\times10^1}$ | $72.9_{\pm0.5}$ | $\mathbf{53.8}_{\pm1.1}$ | $\mathbf{33.4}_{\pm0.6}$ | $\mathbf{54.0}_{\pm0.3}$ |

### A.5.6 EXPERIMENTS ON TABULAR DATA

We considered analyzing tabular data as an experiment for data with completely different characteristics and features from image data. We used Helena, a dataset for classification with 100 classes. Since the data is not divided for validation and test, we randomly extracted 20 samples per class without duplicates. The distribution of the training data is similar to that of long-tailed data with $\rho \simeq 40$. We call the class $k$ *Many* if the number of training samples satisfies $500 < N_k$, *Medium* if the number of training samples fullfills $200 \leq N_k \leq 500$, and *Few* otherwise for this dataset.

Kadra et al. (2021) show that even an MLP with carefully tuned regularization outperforms state-of-the-art models in the classification of tabular data. Following them, we used a 9-layer MLP with 512 dimensions per layer for the feature extractor. The model is trained by AdamW (Loshchilov & Hutter, 2018) for 400 epochs in the first stage and 10 epochs in the second stage. Thus, WD is not implemented in L2 regularization in this dataset, but is built into the optimizer. By a hyperparameter search with the validation data, we set the hyperparameters as follows. We used dropout (Srivastava et al., 2014) and set the hyperparameter to 0.15. The initial learning rate was 0.001 for the first stage and 0.0004 for the second stage. We set $\lambda$ of WD to 0.15 for the first stage and 0.0003 for the second stage. We set $\zeta$ of FR to 0.001.

Table 16 presents the results. As in the experiment with image data, the proposed method outperforms the existing methods in both test FDR and average accuracy. Note that we used AdamW for the optimizer instead of SGD and that the proposed method also succeeds in this case. This result indicates that the proposed method works for optimizers other than SGD.

### A.6 BROADER IMPACTS

Our research provides theoretical and experimental evidence for the effectiveness of an existing ad-hoc method. We also show that the original method can be simplified based on this theory. The theory is valid for general DNNs and does not concern any social issues such as privacy. On the contrary, it reduces the number of training stages to one while maintaining a higher level of accuracy, thus reducing the computational cost and the negative impact on the environment.

Table 16: FDRs and accuracy of MLP trained with each method for Helena.

| Method | LA | FDR | | Accuracy (%) | | | |
|---|---|---|---|---|---|---|---|
| | | Train | Test | *Many* | *Medium* | *Few* | Average |
| CE | N/A | $5.40\times10^1_{\pm5.7\times10^0}$ | $6.48\times10^1_{\pm1.1\times10^0}$ | $34.7_{\pm0.8}$ | $20.8_{\pm1.1}$ | $17.2_{\pm0.5}$ | $24.3_{\pm0.5}$ |
| CB | N/A | $4.31\times10^1_{\pm7.1\times10^0}$ | $6.53\times10^1_{\pm0.7\times10^0}$ | $26.4_{\pm0.8}$ | $27.5_{\pm1.0}$ | $\mathbf{25.1_{\pm1.8}}$ | $26.3_{\pm0.8}$ |
| WD | N/A | $\mathbf{7.61\times10^1_{\pm1.0\times10^0}}$ | $6.53\times10^1_{\pm0.6\times10^0}$ | $\mathbf{36.3_{\pm0.6}}$ | $20.8_{\pm0.4}$ | $16.6_{\pm1.1}$ | $24.7_{\pm0.6}$ |
| | Add | $\mathbf{7.61\times10^1_{\pm1.0\times10^0}}$ | $6.53\times10^1_{\pm0.6\times10^0}$ | $33.4_{\pm0.4}$ | $26.3_{\pm0.6}$ | $24.5_{\pm0.8}$ | $28.1_{\pm0.2}$ |
| | Mult | $\mathbf{7.61\times10^1_{\pm1.0\times10^0}}$ | $6.53\times10^1_{\pm0.6\times10^0}$ | $31.6_{\pm0.3}$ | $27.1_{\pm0.5}$ | $21.9_{\pm1.0}$ | $26.9_{\pm0.3}$ |
| WB | N/A | $\mathbf{7.63\times10^1_{\pm0.8\times10^0}}$ | $6.55\times10^1_{\pm0.7\times10^0}$ | $35.0_{\pm0.5}$ | $26.0_{\pm0.8}$ | $22.7_{\pm1.4}$ | $28.0_{\pm0.5}$ |
| WD&ETF | N/A | $\mathbf{7.73\times10^1_{\pm7.2\times10^0}}$ | $6.86\times10^1_{\pm0.3\times10^0}$ | $\mathbf{36.2_{\pm0.6}}$ | $22.6_{\pm1.1}$ | $17.7_{\pm0.3}$ | $25.6_{\pm0.4}$ |
| | Add | $\mathbf{7.73\times10^1_{\pm7.2\times10^0}}$ | $6.86\times10^1_{\pm0.3\times10^0}$ | $32.6_{\pm0.9}$ | $28.8_{\pm0.6}$ | $25.5_{\pm0.9}$ | $\mathbf{29.0_{\pm0.4}}$ |
| | Mult | $\mathbf{7.73\times10^1_{\pm7.2\times10^0}}$ | $6.86\times10^1_{\pm0.3\times10^0}$ | $32.1_{\pm1.0}$ | $\mathbf{30.1_{\pm0.6}}$ | $25.4_{\pm0.9}$ | $\mathbf{29.2_{\pm0.3}}$ |
| WD&FR &ETF | N/A | $\mathbf{8.49\times10^1_{\pm14.9\times10^0}}$ | $\mathbf{7.04\times10^1_{\pm0.7\times10^0}}$ | $36.1_{\pm1.3}$ | $21.6_{\pm0.6}$ | $17.4_{\pm1.0}$ | $25.2_{\pm0.3}$ |
| | Add | $\mathbf{8.49\times10^1_{\pm14.9\times10^0}}$ | $\mathbf{7.04\times10^1_{\pm0.7\times10^0}}$ | $31.3_{\pm1.0}$ | $28.7_{\pm0.6}$ | $\mathbf{27.5_{\pm0.6}}$ | $\mathbf{29.2_{\pm0.4}}$ |
| | Mult | $\mathbf{8.49\times10^1_{\pm14.9\times10^0}}$ | $\mathbf{7.04\times10^1_{\pm0.7\times10^0}}$ | $31.9_{\pm1.1}$ | $\mathbf{29.6_{\pm0.9}}$ | $25.6_{\pm0.5}$ | $\mathbf{29.1_{\pm0.5}}$ |

