# OpenReview forum: "Exploring Weight Balancing on Long-Tailed Recognition Problem"
_ICLR.cc/2024/Conference — ICLR 2024 poster_

### Official Review · Reviewer_2xQy · 2023-10-19

**Soundness:** 3 good
**Presentation:** 3 good
**Contribution:** 3 good
**Rating:** 6
**Confidence:** 3

**Summary:**

The paper aims to analyze weight balancing by examining neural collapse and the cone effect at each training stage. The analysis reveals that weight balancing can be broken down into an increase in Fisher's discriminant ratio of the feature extractor due to weight decay and cross entropy loss, as well as implicit logit adjustment caused by weight decay and class-balanced loss. This analysis allows for a simplified training method with only one training stage, while improving accuracy.

**Strengths:**

1. As an experimental and analytical paper, the logical flow of the entire article is smooth, providing a good reading experience.
2. Weight Decay, as a simple yet effective model, is thoroughly explained in this paper with targeted explorations and explanations at each step. The argumentation is well-grounded and convincing.
3. The feasibility of single-stage training is explored based on the analysis of the original method, which represents a certain breakthrough.

**Weaknesses:**

1. The analysis solely based on one particular model method has certain limitations, as it lacks consideration of other methods. Exploring why Weight Decay performs exceptionally well indeed raises a thought-provoking question in the long-tail domain. However, the favorable properties of Weight Decay have already been extensively explored in balancing datasets, and its effectiveness can be considered widely recognized.

**Questions:**

1. Besides the analysis metrics mentioned in the paper, what other commonly used metrics exist? Why was the choice of metrics in the paper considered?
2. If we consider balanced datasets, the analysis in the paper can still hold true. The only difference lies in the performance based on the sota models. What distinguishes this type of analysis from conventional methods when dealing with long-tailed data distributions? What are the innovative aspects of this paper?

---

> ### Author Response · Authors · 2023-11-16
>
> We sincerely appreciate your time and thoughtful evaluation of our paper. Your positive feedback is genuinely encouraging, and we are grateful for the constructive comments that have undoubtedly contributed to the improvement of our work.
>
> While we understand and value the favorable assessment, we would like to address a few points raised in your review where I believe additional context or clarification could potentially enhance the overall evaluation of our paper.
>
>
> > W1: The analysis solely based on one particular model method has certain limitations, as it lacks consideration of other methods. Exploring why Weight Decay performs exceptionally well indeed raises a thought-provoking question in the long-tail domain. However, the favorable properties of Weight Decay have already been extensively explored in balancing datasets, and its effectiveness can be considered widely recognized.
>
> A: We believe that our research, through the analysis of WB, will contribute more than just understanding one method. Our research has two major strengths: 1. we have proved that WD prevents the cone effect [1], a phenomenon that is undiscovered; 2. we have identified fundamental elements that are useful not only for WB but also for general LTR methods. The cone effect is a recently discovered phenomenon, which is meaningful but little-studied. This effect is an inductive bias that is common in deep neural networks, where the output features tend to have high cosine similarity. It is an obstacle in training classification models since low inter-class cosine similarity is desirable. Theorem 1 mathematically shows that regardless of the distribution of the training data, training with WD and CE prevents the increase in inter-class cosine similarity. This is a significant result that suggests an effective way to train classification models not only for LTRs.
>
>  Our study is also meaningful because it is not only an extension of WB but breaks WB down into the fundamental elements important in LTR methods. WB is a simple but effective method that has attracted attention as a baseline for LTR. We analyze this method both theoretically and experimentally to identify the basic elements that are useful for LTR training. Thus, our study is also significant enough to serve as a basis for future research. Such a simple algorithm with theoretical analysis for the long-tail problem has a useful effect on the entire application field in deep learning because of its simplicity.
>
>
> > Q1:  Besides the analysis metrics mentioned in the paper, what other commonly used metrics exist? Why was the choice of metrics in the paper considered?
>
> A: We have chosen the evaluation metrics in terms of eliminating arbitrariness as much as possible. We have followed existing studies and selected accuracy. We have adopted the accuracy within each group when classes were categorized by the number of samples and the average accuracy between classes. These metrics are commonly used in LTR studies such as [2] and [3]. We have adopted FDR as a metric for the ease of linear feature separation. Other metrics include the accuracy of unsupervised learning models, e.g., the nearest neighbor search method [2]. However, this metric is arbitrary in terms of which unsupervised learning method we use and the hyperparameters we set. FDR has the advantage of being free from such arbitrariness and relatively simple to compute.
>
>
> > Q2: If we consider balanced datasets, the analysis in the paper can still hold true. The only difference lies in the performance based on the sota models. What distinguishes this type of analysis from conventional methods when dealing with long-tailed data distributions? What are the innovative aspects of this paper?
>
> A:Our study is innovative in that it demonstrates that LTR is not a special setting. As shown in Table 8 in the Appendix, many existing studies assume that LTR is an unusual setting and improve the accuracy by applying complex techniques. However, the reality is that LTRs are not so extraordinary and that simple methods that can be used to balanced data are sufficient to improve accuracy. The fact that our theory and experiments gave the impression that our analysis is no different from that of the balanced case, which is precisely the innovative point.

---

> > ### Author Response · Authors · 2023-11-16
> >
> > In responding to your feedback, our goal is to not only meet the criteria for acceptance but also to exceed expectations where possible. We believe that by addressing these points, we can enhance the overall contribution and impact of our paper. We genuinely appreciate your careful consideration of our work, and we are committed to ensuring that the revised manuscript reflects the highest standards of quality and clarity. If you have any additional suggestions or concerns, please don't hesitate to let us know.
> >
> > Thank you once again for your time and valuable insights.
> >
> > [1] Mind the Gap: Understanding the Modality Gap in Multi-modal Contrastive Representation Learning, Liang+, NeurIPS2022
> >
> > [2] Decoupling Representation and Classifier for Long-Tailed Recognition, Kang+, ICLR2020
> >
> > [3] Long-tail learning via logit adjustment, Menon+, ICLR2021

---

> > > ### Comment · Reviewer_2xQy · 2023-11-18
> > >
> > > Thank you for the response, which has helped resolve my confusion quite well. I will maintain my current score as positive feedback.

---

### Official Review · Reviewer_GF5W · 2023-10-27

**Soundness:** 3 good
**Presentation:** 3 good
**Contribution:** 3 good
**Rating:** 6
**Confidence:** 4

**Summary:**

This paper primarily investigates why the two-stage WD method could perform well in long-tailed tasks. It analyzes the WB by focusing on
neural collapse and the cone effect at each training stage and found that it can be decomposed into an increase in Fisher’s discriminant ratio of the feature extractor caused by weight decay and cross-entropy loss and implicit logit adjustment caused by weight decay and class-balanced loss. Then the paper proposes the simplify the WD by reducing the number of training stages into one with the combination of WD, FR, and ETF.

**Strengths:**

1. This paper provides an in-depth analysis of the reasons behind the success of WD in long-tail scenarios, demonstrating thoughtful insights. From the perspective of neural collapse and the cone effect, it explains the WD well.
2. This paper has a well-organized structure which makes it easy for readers to understand the research.
3. Extensive experimental results confirm the validity of the analysis.

**Weaknesses:**

1. The paper only discusses the related work of NC and WD but the related work of the long-tail is also necessary.
2. Some concerns which I will mention in the following section.

**Questions:**

1. What's the meaning of O in Eq.3 and could the author explain more about Theorem 2?
2. Could the author explain why the WD&FR&ETF performs worse than the WD&ETF on the ImageNet-LT dataset in Table 13? And are there any experiments conducted on large-scale datasets, such as iNaturalist 2018?
3. Existing long-tail solutions often rely on expert systems to improve the performance of tail classes, such as RIDE[1] and SADE[2]. Is the proposed method in this paper compatible with them?

[1] Wang, Xudong, et al. "Long-tailed recognition by routing diverse distribution-aware experts." arXiv preprint arXiv:2010.01809 (2020).
[2] Zhang, Yifan, et al. "Test-agnostic long-tailed recognition by test-time aggregating diverse experts with self-supervision." arXiv e-prints (2021): arXiv-2107.

---

> ### Author Response · Authors · 2023-11-16
>
> We sincerely appreciate your thorough review of our paper and your positive feedback. Your insights have been invaluable in shaping the direction of our work. We are grateful for the time and effort you dedicated to providing constructive comments.
> We would like to address some of the points raised in your review and provide additional context or clarification that we believe could contribute to an even more favorable evaluation. We indicate revised sections in red in the PDF.
>
>
> > W1: The paper only discusses the related work of NC and WD but the related work of the long-tail is also necessary.
>
> A1: We have added an overview of LTR-related methods in Appendix A.2. We put it in Appendix because of the strict page limit of the main text and because we have already briefly introduced the LTR methods in Introduction. In addition, we have added the following at the end of the first sentence of the second paragraph of Section 1 to make it clear that this section is present.
> - see Appendix A.2 for more related research.
>
>
>
> > Q1:  What's the meaning of O in Eq.3 and could the author explain more about Theorem 2?
>
> A: This is Bachmann-Landau O-notation, and Theorem 2 indicates that the second stage of WB operates equivalent to multiplicative LA. The notation $O\left(\frac{1}{x}\right)$ indicates that the term is at most a constant multiple of $\frac{1}{x}$ if $x$ is sufficiently large. For example, in the second stage of WB for CIFAR100-LT in [1], $\lambda \rho C = 1000 \gg 1$. Often in the text, we have used a notation that uses O notation and inequality, but since this does not seem very common, we have changed it to a notation of equal signs. Also, to clarify the meaning of $O$, the following sentence is added after Theorem 2.
> - where $O$ is a Bachmann-Landau O-notation. For example, the notation in the form $O\left(\frac{1}{x}\right)$ indicates that the term is at most a constant multiple of $\frac{1}{x}$ if $x$ is sufficiently large.
>
> Theorem 2 means that in the second stage of WB, the linear layer is trained so that the logit is larger for tail classes. If $\lambda \rho C$ is large enough, then $\mathbf{w}_{k}^* \sim \frac{\overline{N}}{\lambda N} \boldsymbol{\mu}_k$ is satisfied. Note that the features are trained to have larger norm for tail classes in the first stage of training. Therefore, in the second stage of training, each row vector in the linear layer is oriented in the same direction as the corresponding class feature, and the norm is higher for tail classes. This operation is equivalent to multiplicative LA. See also Section 5.1.
>
>
>
> > Q2: Could the author explain why the WD&FR&ETF performs worse than the WD&ETF on the ImageNet-LT dataset in Table 13? And are there any experiments conducted on large-scale datasets, such as iNaturalist 2018?
>
> A: FDR is improved by FR also in the results of the ImageNet-LT experiment. The purpose of adding FR is to make it easier to satisfy the assumption of Theorem 1 and to obtain features that are more easily linearly separable. In this respect, the experimental results are consistent with the theory. On the other hand, FR does not significantly degrade accuracy, but fails to improve. We find that FR begins to have an effect when the training samples are almost completely separated. Figure 1 in the supplementary material shows the relationship between the number of training epochs, training accuracy, and FDR for CIFAR100-LT. After 200-epoch training, the difference between training with and without FR increases. At this point, the training accuracy is almost 100%. This suggests that FR alone may fail to have a significant impact on test accuracy itself.
>
>
> Experiments on other large data sets are considered to be quite difficult due to the lack of an experimental environment in our university. For example, a typical experiment on iNaturalist in LTR requires training with a batch size of 500, which requires a lot of VRAM or GPU hours [1]. In addition, although WB publishes the results of its iNaturalist experiments, it does not disclose the hyperparameters used in the training. We need to explore them for comparison, which requires more computational resources. The cloud servers we use for our experiments have a limited number of GPUs that can be used simultaneously for long periods of time, making the experiments difficult. We also considered cloud computing services such as AWS, but this is also difficult due to its enormous cost. On the basis of our experimental environment for ImageNet-LT, we would need to use a p3dn.24xlarge instance for more than 100 hours on AWS, for example. The on-demand fee for this is about $3100, which is too expensive for this supplemental experimentation. For these reasons, we have determined that the ImageNet-LT results are sufficient.

---

> > ### Author Response · Authors · 2023-11-16
> >
> > > Q3: Existing long-tail solutions often rely on expert systems to improve the performance of tail classes, such as RIDE[1] and SADE[2]. Is the proposed method in this paper compatible with them?
> >
> > A: Yes, the proposed combination is compatible with these methods because of its simplicity.
> >
> >
> > In making these revisions, we believe the paper has achieved a higher level of rigor and clarity. We hope these changes address your concerns and align with your expectations for an even more impactful contribution to the field.
> > Should you have any further suggestions or specific areas you believe could benefit from additional attention, we welcome your guidance. Your expertise is invaluable to us, and we are committed to delivering a paper that meets the highest standards.
> >
> > Thank you once again for your time and thoughtful feedback. We look forward to the opportunity to further refine our work based on your insights.
> >
> > [1] Long-Tailed Recognition via Weight Balancing, Alshammari+, CVPR2022
> >
> > [2] Self-Supervised Aggregation of Diverse Experts for Test-Agnostic Long-Tailed Recognition, Zhang+, NeurIPS2022

---

### Official Review · Reviewer_7mhz · 2023-10-27

**Soundness:** 4 excellent
**Presentation:** 3 good
**Contribution:** 4 excellent
**Rating:** 8
**Confidence:** 3

**Summary:**

This paper studies the problem of long-tailed recognition (LTR) and presents theoretical analysis regarding the two-stage training of LTR. The main findings include two theorems showing 1) how neural collapse and the cone effect are affected by weight balancing at each training stage; 2) how weight decay contributes to an increased in Fisher's discriminant ratio of the feature extractor and implicit logit adjustment. In addition to those theoretical results, authors also report extensive experimental results as supporting evidence. The paper is well-written and easy to follow. The technical contributions of this work are expected to sharpen our understanding of the LTR problem, which might inspire other attacks to LTR than weight balancing.

**Strengths:**

1. The problem formulation is well motivated and sensible. Developing a theory for weight balancing in LTR has been under-researched in the literature. This work makes a timely contribution to this important topic.
2. The technical contributions in Sec. 4 and 5 are solid. Both theorems 1 and 2 are well presented and their rigorous proof have been included in the Appendix. The generalized result of Theorem 2 (Theorem 3 in Appendix) is commendable.
3. In addition to the theoretical analysis, this paper also reported extensive experimental results as supporting evidence. Those figures and tables have greatly facilitated the understanding of the underlying theory.

**Weaknesses:**

1. The difference between weight balancing (WB) and weight decay (WD) needs to be make clearer. Sec. 3 only reviews WB and overlooks WD. Historically, WD was proposed much earlier than WB. It will be a good idea to include some review of WD in Sec. 3, I think. Note that WD is already present in Table 1 on page 4 (right after Sec. 3).
2. For those who are less familiar with two-stage training of LTR, it might be a good idea to include a concise review of two-stage training methods in the Appendix. Note that CVPR2022 and ICLR2020 have different formulation of two-stage training. Please clarify that the model analyzed in this paper is based on the CVPR2022 work even though it cited the ICLR2020 as the original source of two-stage training.
3. There are many acronyms in this paper. It might be a good idea to provide a table summarizing them in the Appendix A.1 (Notation and Acronym).

**Questions:**

1. What do blue and red colors in Table 4 and Table 9 highlight? Some explanations can be added to the caption of those tables.
2. Table 1 includes experimental results for WD without and with fixed batch normalization (BN). Any plausible explanation for these results? Why does BN further improve the performance of WD?
3. In Table 5, the accuracy performance of LA (N/A) is noticeably higher than add/mult for the category of "many". Why does LA only work for the Medium and Few classes?

---

> ### Author Response · Authors · 2023-11-14
>
> We extend our sincere gratitude for your meticulous review of our paper. We are truly honored and encouraged by the positive evaluation and constructive feedback you provided. Your insights have been invaluable in refining our work.
>
> We are pleased to note that our research has met with your approval. Your comments have reaffirmed our commitment to maintaining the highest standards of academic rigor. We have carefully considered your suggestions and have made the following revisions to further enhance the clarity and impact of our paper. Revised sections are indicated in red in the PDF.
>
> > W1: The difference between weight balancing (WB) and weight decay (WD) needs to be make clearer. Sec. 3 only reviews WB and overlooks WD. Historically, WD was proposed much earlier than WB. It will be a good idea to include some review of WD in Sec. 3, I think. Note that WD is already present in Table 1 on page 4 (right after Sec. 3).
>
> A: We have added the following sentence to the beginning of the paragraph "Regularization Methods" in Section 3.
> - We implemented WD as L2 regularization with a hyperparameter $\lambda$ because we used stochastic gradient descent (SGD) for the optimizer. Thus, optimization is written as $\mathbf{\Theta}^* = \arg\min_{\mathbf{\Theta}}F(\mathbf{\Theta}; \mathcal{D}) +  \frac{\lambda}{2}\sum_{k \in \mathcal{Y}}\|\mathbf{w}_{k}\|_2^2$.
>
> In addition, we have deleted the following sentence from the paragraph "Weight Balancing" in Section 3 due to duplication.
> -  "Note that we implemented WD as L2 regularization with a hyperparameter λ because we used stochastic gradient descent (SGD) for the optimizer."
>
>
>
> > W2: For those who are less familiar with two-stage training of LTR, it might be a good idea to include a concise review of two-stage training methods in the Appendix. Note that CVPR2022 and ICLR2020 have different formulation of two-stage training. Please clarify that the model analyzed in this paper is based on the CVPR2022 work even though it cited the ICLR2020 as the original source of two-stage training.
>
> A: We have added an overview of LTR-related methods and two-stage learning in Appendix A.2. For example, the section on two-stage learning is as follows. Please refer to the PDF for details.
> - Two-stage learning (Kang et al., 2020) is a method for improving accuracy by dividing LTR training into two stages: feature extractor training and classifier training. It is used in numerous methods (Cao et al., 2019; Ma et al., 2021; 2022; Li et al., 2022; Tian et al., 2022; Liu et al., 2023; Kang et al., 2023), including WB, because of its simple but significant improvement in accuracy. Note that since our work analyzes WB, the formulation of two-stage learning is based on WB. For example, the classifier weights are initialized randomly at the start of the second stage of training in Kang et al. (2020) but not in WB.
>
>
> > W3. There are many acronyms in this paper. It might be a good idea to provide a table summarizing them in the Appendix A.1 (Notation and Acronym).
>
> A: We have added a table of abbreviations in Appendix A.1. Thank you for your constructive suggestion!
>
> > Q1: What do blue and red colors in Table 4 and Table 9 highlight? Some explanations can be added to the caption of those tables.
>
>
> A: Red text indicates that the FDR increases when learned features pass through an additional layer, and blue text indicates that it decreases. In Table 4, when the after FDR is higher than the before, and in Table 9 (Table 10 in the modified version), when the FDR is higher than the FDR with one less additional layer passed, the values are marked in red.
> In this section, we examine whether the features learned by each method are more linearly separable when pass through a randomly initialized linear layer and ReLU. Therefore, the difference in FDR before and after passing an additional layer is significant. To explicitly indicate this purpose, we have add the following sentence to each caption.
> - Table 4
> 	- Red text indicates higher values than before, blue text indicates lower values.
> - Table 9 (10)
> 	- Red text indicates an increase compared to the FDR of the feature when the number of layers passed is one less, while blue text indicates a decrease.

---

> > ### Author Response · Authors · 2023-11-14
> >
> > > Q2: Table 1 includes experimental results for WD without and with fixed batch normalization (BN). Any plausible explanation for these results? Why does BN further improve the performance of WD?
> >
> > A: We discuss positive effect on applying WD to BN in Section 4.3 and Appendix A.5.2. "WD w/o BN" represents when WD is applied to other than the BN layers. Note that BN is used in the model. In contrast, "WD fixed BN" indicates when the scaling parameter in the BN layer is fixed to a common small value and the model is trained with WD. Comparing "WD w/o BN" with "CE w/ WD" and "WD fixed BN", only "WD w/o BN" results in a lower FDR. This result suggests that WD facilitates learning by reducing the scaling parameters of the BNs. For example, the smaller scaling of the BN in the final layer decreases the norm of the features and makes it easier to satisfy the assumptions of Theorem 1. In Appendix A.5.2, we discuss experiments that suggest applying WD may positively affect learning by increasing the variance of the shifting parameters of the BN compared to the mean.
> >
> >
> >
> > > Q3: In Table 5, the accuracy performance of LA (N/A) is noticeably higher than add/mult for the category of "many". Why does LA only work for the Medium and Few classes?
> >
> > A: Because LA increases the probability of being classified into tail classes. LA is a method to correct classification bias by manipulating the output logit for each class. For example, Additive LA adds larger values for classes with fewer samples, while Multiplicative LA multiplies. As a result, LA increases the probability of being classified into tail classes and the accuracy of the classes.
> >
> >
> > We believe that these revisions contribute positively to the overall strength of our paper, and we hope they align with your expectations.
> > We sincerely appreciate the time and effort you dedicated to the review process. Your positive feedback serves as motivation to continually improve our research. If you have any additional suggestions or comments to ensure the paper's excellence, we are more than willing to incorporate them.
> > Thank you once again for your generous evaluation and constructive feedback. We are excited about the opportunity to contribute this work to the academic community.

---

### Official Review · Reviewer_dLbF · 2023-10-30

**Soundness:** 2 fair
**Presentation:** 2 fair
**Contribution:** 2 fair
**Rating:** 6
**Confidence:** 4

**Summary:**

The author analyzed the weight balancing method for long-tailed classification problems from the perspectives of neural collapse and the cone effect, and provided some insights.

**Strengths:**

1. The problem of imbalanced classification is undeniably a highly practical and crucial research issue in the field of machine learning.
2. The authors provided an analysis of weight balancing to a certain extent and offered insightful perspectives on the topic.

**Weaknesses:**

1. This paper appears to resemble an appendix on Weight Balancing to some extent and the technical innovation is rather limited.
2. Given that Weight Balancing is not the best-performing method in the field of imbalanced learning, the significance of this paper in the field remains debatable.
3. Considering that Weight Balancing involves implicit constraints at the parameter level (compared to direct correction in other long-tail classification methods), its extension to address broader distribution shift issues should hold greater value.
4. Sec 5.1"the second stage of WB is equivalent to multiplicative LA". Why not just use explicit LA?


update: After reading the authors' response and other reviewers' comments, I would like to increase my score to weak accept.

**Questions:**

See above

---

> ### Author Response · Authors · 2023-11-14
>
> We sincerely appreciate the time and effort you dedicated to reviewing our paper. While we are grateful for your constructive comments, we also note your observation that our work might not quite meet the acceptance criteria. We would like to address your concerns and present our perspective on these points.
>
> > W1: This paper appears to resemble an appendix on Weight Balancing to some extent and the technical innovation is rather limited.
>
> A1: Our work differs from [1] in the following points: 1. we compare FDRs; 2. we provide theoretical bases; 3.  we demonstrate and experiment with more effective combinations of methods based on those bases.
> Indeed, while the Appendix in [1] compares the accuracy of various combinations of regularization methods, we compared FDRs to quantify the ease of linear separation of features. We also present a theory consistent with these experimental results, confirming a theoretical foundation not presented in [1]. On these grounds, we also present combinations not considered in [1], such as ETF, LA, and FR, and demonstrate their validity. In these respects, our work is more than just an extension of [1].
>
>
> > W2: Given that Weight Balancing is not the best-performing method in the field of imbalanced learning, the significance of this paper in the field remains debatable.
>
> A2:  WB is an essential method suitable for baselines and that theoretical analysis and improvement of it is meaningful. As shown in [1] and Table 7 (Table 8 in the modified version) in the Appendix of our paper, models that have achieved SOTA, such as [2], are complex methods combining ensembles, etc. However, with minimal effort, WB achieves higher accuracy than other existing methods [1]. Therefore, [1] is considered important as a baseline and has attracted attention, with more than 70 citations. We analyze this method both theoretically and experimentally to identify the basic elements that are useful for LTR training. Thus, our study is also significant enough to serve as a basis for future research.
>
> Our research also reveals significant properties of DNN training in general, not just the analysis of WB. It is also credited with discovering the new importance of WD and CE in training classification models, where WD and CE prevent the cone effect [3]. This phenomenon is an inductive bias that is common in deep neural networks, where the output features tend to have high cosine similarity. It is an obstacle in training classification models since low inter-class cosine similarity is desirable. Theorem 1 mathematically shows that regardless of the distribution of the training data, training with WD and CE prevents the increase in inter-class cosine similarity. This is a significant result that suggests an effective way to train classification models not only for LTRs.
>
>
> > W3: Considering that Weight Balancing involves implicit constraints at the parameter level (compared to direct correction in other long-tail classification methods), its extension to address broader distribution shift issues should hold greater value.
>
> A3: Problems dealing with shifts to uniform distributions are highly significant. Many studies dealing with shifts to uniform distributions have been published, including [1] and [2], which have received attention. Indeed, addressing general distributional shifts, such as [4], has also attracted interest. However, addressing shifts to uniform distributions has become more significant. For example, [4] trains one neural network of an ensemble using existing methods for shifts to uniform distributions. Thus, the methods for shifts to uniform distributions are the basis of methods for general shifting distributions, and improvements to them are of sufficient importance.

---

> ### Author Response · Authors · 2023-11-14
>
> > W4: Sec 5.1"the second stage of WB is equivalent to multiplicative LA". Why not just use explicit LA?
>
> A4: We verify the effectiveness of replacing the second stage in WB with LA in Section 5.2. We analyze the effectiveness of WB both experimentally and theoretically before Section 5.2. In Section 5.2, we explicitly replace WB with essential elements to test the validity of the combination. We also test the case where we explicitly replace the second stage of training with LA. Table 5 shows that the appropriate combination including LA achieves better accuracy than WB. We have changed the final sentence of Section 5.1 as follows to clarify this logic. See the modified PDF for details. Updated sections in the PDF are indicated in red.
> - This also suggests that replacing the second stage of training with LA would be more generic and we confirm its validity in the following section.
>
>
> In light of these revisions, we kindly request a reevaluation of our paper for acceptance. We are confident that the changes made address the concerns raised, and the paper now meets the high standards set.
> If there are specific aspects that still fall short of expectations, we would greatly appreciate further guidance on how we can refine our work to meet the standards for acceptance.
> Thank you for your thorough review and the opportunity to enhance our paper. We are committed to making the necessary revisions to ensure the highest quality of research.
> We look forward to your further feedback and hope that the revised version of our paper will meet the standards for acceptance.
>
> [1] Long-Tailed Recognition via Weight Balancing, Alshammari+, CVPR2022
>
> [2] Retrieval Augmented Classification for Long-Tail Visual Recognition, Long+, CVPR2022
>
> [3] Mind the Gap: Understanding the Modality Gap in Multi-modal Contrastive Representation Learning, Liang+, NeurIPS2022
>
> [4] Self-Supervised Aggregation of Diverse Experts for Test-Agnostic Long-Tailed Recognition, Zhang+, NeurIPS2022

---

> ### Comment · Reviewer_dLbF · 2023-11-14
>
> After reading the authors' response and other reviewers' comments, I would like to increase my score to weak accept.

---

> > ### Author Response · Authors · 2023-11-14
> >
> > Thank you for your reevaluation! Your positive evaluation in our work are genuinely appreciated. We are pleased to learn that our efforts in this regard have resonated with you.

---

### Comment · Area_Chair_V2me · 2023-11-17
**Author-Reviewer Discussion Phase**

Thank you, reviewers, for your work in evaluating this submission. The reviewer-author discussion phase takes place from Nov 10-22.

If you have any remaining questions or comments regarding the rebuttal or the responses, please express them now. At the very least, please acknowledge that you have read the authors' response to your review.

Thank you, everyone, for contributing to a fruitful, constructive, and respectful review process.

AC

---

### Meta-Review · Area_Chair_V2me · 2023-12-05

**Metareview:**

This paper revisits the weight balancing (WB) method for the long-tailed recognition problem. It analyzes the effect of WB from the perspectives of neural collapse and cone effect, and finds that the weight decay (WD) in WB can contribute to an increase in Fisher's discriminant ratio of the feature extractor, as well as implicit logit adjustment. Both theoretical and empirical analyses have been conducted to verify the idea. Moreover, the authors have properly organized the paper to provide a clear and in-depth understanding of how WB helps.
During the rebuttal phase, the authors have properly responded to the reviewers' comments, including the concerns about limited contribution and the need for more explanations. After the rebuttal, all of the reviewers made positive comments. Therefore, I recommend the acceptance of this paper.

**Justification For Why Not Higher Score:**

This work mainly studies the effect of weight balancing (WB) in long-tailed recognition. Since WB is a two-stage approach, the studied problem may be not thorough enough. In long-tailed recognitions, there are various methods under the single-stage framework. Also, there are some methods using ensemble approaches or more complex backbones. Therefore, I suggest the authors extend their work to more situations in the future.

**Justification For Why Not Lower Score:**

This paper systematically investigates the reason behind the effectiveness of weight balancing (WB) for long-tailed recognition. Both theoretical and empirical studies are conducted. The results provide an interesting insight, i.e., the weight decay (WD) can yield a significant influence in two-stage frameworks. Therefore, this work provides an interesting insight and can inspire future research for long-tailed learning.

---

### Decision · Program_Chairs · 2024-01-16

Accept (poster)